# Digital Innovation Enabled Nanomaterial Manufacturing; Machine Learning Strategies and Green Perspectives

**DOI:** 10.3390/nano12152646

**Published:** 2022-08-01

**Authors:** Georgios Konstantopoulos, Elias P. Koumoulos, Costas A. Charitidis

**Affiliations:** 1RNANO Lab—Research Unit of Advanced, Composite, Nano Materials & Nanotechnology, School of Chemical Engineering, National Technical University of Athens, GR15773 Athens, Greece; gkonstanto@chemeng.ntua.gr (G.K.); charitidis@chemeng.ntua.gr (C.A.C.); 2Innovation in Research & Engineering Solutions (IRES), Boulevard Edmond Machtens 79/22, 1080 Brussels, Belgium

**Keywords:** nanomaterials, artificial intelligence, machine learning, in silico design of materials, data-driven engineering, manufacturing

## Abstract

Machine learning has been an emerging scientific field serving the modern multidisciplinary needs in the Materials Science and Manufacturing sector. The taxonomy and mapping of nanomaterial properties based on data analytics is going to ensure safe and green manufacturing with consciousness raised on effective resource management. The utilization of predictive modelling tools empowered with artificial intelligence (AI) has proposed novel paths in materials discovery and optimization, while it can further stimulate the cutting-edge and data-driven design of a tailored behavioral profile of nanomaterials to serve the special needs of application environments. The previous knowledge of the physics and mathematical representation of material behaviors, as well as the utilization of already generated testing data, received specific attention by scientists. However, the exploration of available information is not always manageable, and machine intelligence can efficiently (computational resources, time) meet this challenge via high-throughput multidimensional search exploration capabilities. Moreover, the modelling of bio-chemical interactions with the environment and living organisms has been demonstrated to connect chemical structure with acute or tolerable effects upon exposure. Thus, in this review, a summary of recent computational developments is provided with the aim to cover excelling research and present challenges towards unbiased, decentralized, and data-driven decision-making, in relation to increased impact in the field of advanced nanomaterials manufacturing and nanoinformatics, and to indicate the steps required to realize rapid, safe, and circular-by-design nanomaterials.

## 1. Introduction

The modern needs of science move towards sustainable design and the re-use of materials in order to expand the life-span, reduce the environmental impact of the composite components, as well as to use resources wisely. More specifically, ontology-assisted product lifecycle management has stepped up to support the materials design and manufacturing, usage and maintenance, and end of life management including recycling, disposal, and reuse [1,2]. Specifically, the reuse of materials can maximize the benefit at societal and environmental levels via the utilization of the excess of carbon dioxide [3,4]. Moreover, Nanotechnology plays a leading role in advancing technologies in several sectors from the natural sciences to interdisciplinary engineering fields, including a societal impact by the creation of new and attractive job positions, the financial growth, and the improved quality of life. It is estimated that by the end of 2022, the total market value of nanomaterials will exceed $90.5 billion worldwide, while the demand for commercial and consumer nano-enhanced products demonstrated a growing trend in recent years [5].

Modern societal and industrial needs require proper management of the produced data to address the ambition and evolution of Industry 4.0 and human-oriented Industry 5.0 program in Europe, Society 5.0 program in Japan, and Made in EU 2025 program in China [6,7,8]. Nanoinformatics field growth has been more challenging than its predecessors (cheminformatics, materials informatics), requiring multidisciplinary scientific teams to solve multiscale problems with many variables affecting the structure–property relations. In order to monitor nanomaterial data in real-time, a demanding infrastructure is required, including functional devices with sensors and/or actuators, which provide an interface with machine intelligence [9]. These systems are expected to serve real-world applications in the near future, such as buildings, vehicles, and even whole cities, which will communicate within a power grid. Another field of nanoinformatics and machine intelligence expected to flourish is the wearable electronics to support biomedical and medicine research where lot of data are produced, e.g., human–machine interfaces, on-body biosensors, and artificial skins, as well as the fields of medical diagnosis and monitoring of health will benefit from those technologies [10]. For instance, a smart tattoo, smart bandage, and clothing biosensors have already demonstrated proof-of-concept detection of biomarkers, such as lactase, glucose, electrolyte ions, alcohol level, hydration status, and for the evaluation of, e.g., wound status to prevent potential local infection [11].

In the field of nanomanufacturing, the establishment of novel predictive models is challenging and requires scalable, fast, and accurate data acquisition to increase the impact in the production. One common limitation is related to data or failed results which are not accessible or methods applied may suffer from irreproducibility [10], and this is evidenced when researchers publish or reproduce a work with contradictory/different outcomes. This is attributed to a lack of documentation, while the failed results instead of being lost can be key for AI to enhance the interpretation of algorithms, i.e., establish cost factors/criteria and optimize next experiments. As it is indicated, the need for standardized, curated, and accessible data is emerging to guide informed predictions [12].

Especially, progress in data-driven manufacturing and evidence-based decision support systems includes digital metrology for the real-time assessment of the production quality. Ontology can be the engine to organize and connect the process and materials metadata, which is a key enabler for AI to realize its prospect. Currently, MASON ontology has been used for establishing semantics, towards zero-defect manufacturing [13,14]. The use of ontology-assisted co-simulation and cognitive digital twin is a main driver for the future of manufacturing utilizing real-time data for status monitoring, fault diagnosis, and performance prediction [15]. The need for standardization and semantics has been highlighted in the materials characterization field, and data exchange procedures have also been major topics of workshops and forums of experts [3]. A characterization data structure, “CHADA”, has been developed for the proper documentation of (characterization) experiments facilitating the organization of data, in cooperation with standardization organisations (CEN). “CHADA” is going to be further developed under the umbrella of European Materials Characterization Council (EMCC) to provide interoperability by developing EMMO-compliant domain ontologies [16]. Interoperability was also the dogma in the Materials Genome Initiative, as well as doing, recording, and sharing science [3,17]. In this way, the challenge of high-quality data acquisition suitable for use in predictive modelling tools can be addressed [18]. The European Materials Modelling Council (EMMC) published in their Roadmap the orientations in modelling and AI activities regarding the process optimization and the engineering of (nano-)materials, which require structured data to maximise their impact and support the European industry transition in accordance with digitalization and Green Deal policies [19]. Moreover, the Nanoinformatics Roadmap 2030 focuses on the coordination of data capture, preservation, and dissemination in order to facilitate the accessibility of open data on nanomaterials and envisages structuring databases which will be continuously enriched with data which can be used by computational modelers for establishing structure–property/activity/toxicity relationships as well as to enhance relevant regulatory actions [5,20].

Nowadays, several modelling approaches use known theoretical principles and theories, with the most common methods being the classic molecular dynamics (MD) simulations, density functional theory (DFT), Monte Carlo, finite element analysis (FEM), the phase field method, and the Boltzmann equation of transport [21], which have often been used to model the mechanical properties and thermal conductivity [22]. The use of experimental data in these methods is often limited to the input parameters (also due to time-constraints), while one advantage of machine learning is that the output parameters can feed in addition the same or other machine learning models. Although simulation seems to limitations, AI can deliver a robust approach for combining scientific excellence and sustainability, due to the capacity to utilize data in large-scale and reduce the bias in the prediction of materials behavior, impact, and properties by identifying unseen patterns, while models from chemistry, physics, biology, etc. can be included to increase the precision and reduce variation from batch to batch in a closed-loop laboratory [23].

Relevant progress in computational science has been a key enabler for advancing the Edisonian approach for materials development, which often embraces high cost, risk, and time for the computation/experiment iterations [24]. Understanding nanomaterials in depth has been the motivation to collect key materials descriptors to support the universality of the computational models [25]. Tensile strength, elongation at break, thermal and electrical conductivity, Young’s modulus, porosity, (eco-)toxicity indexes [10,26], topology, crystallography, aspect ratio, zeta potential, surface energy, chemical structure, and protein corona fingerprint have been common descriptors used to establish structure–property–response relationships [27,28,29]. Coming with a single and robust model challenging [28], and requires time investment to collect, curate, and analyze the data. This represents a unique challenge for the future of predictive modelling and nanoinformatics, which raises the degree of complexity for feature extraction, processing, and engineering [30]. In this direction and towards the rational design of materials via data-driven tools, several initiatives have come to provide the most prominent, the U. S. Materials Genome Initiative [3,17].

Regarding model development, the best-case scenario is to have a verified model concerning the correctness and reliability, and to focus on the validation of the model predictions based on the training dataset, in order to avoid high levels of complexity and possible overfitting of the derived model [12,17,23,31,32]. Cross-validation is a common procedure to optimize the prediction metrics and achieve similar fitting accuracy to unseen datasets, but it should be noted that sampling is important, as both the training and validation datasets should be representative of the data [27,32,33,34]. The repeated learning test and k-fold cross-validation have been shown to be the most promising approach to reduce the computational costs in this matter [35]. Consequently, the digitalization of processes and their characterization represent a key ingredient for the success of addressing the interoperability requirements in order to have usable and comparable data, and to accomplish the ambition of a closed-loop laboratory (Figure 1) for the design and selection of suitable (nano)materials [3,23].

Beyond process control, AI and machine learning hold a lot of promise to bridge this gap with their empirical character, as till today rapid characterization tools have been exploited with high accuracy for the discovery of materials and immunotherapies, materials reinforcement/failure/toxicity mechanism recognition, structural characterization, phase detection and quantification, and anomaly detection [3,25,31,33,34,36,37,38,39,40,41,42,43]. While machine learning and AI are already well established in the fields of statistics, economics, and bioinformatics, their utilization in nanotechnology is relatively new [44].

Another concern when using computational methods is the quality of data generated by characterization, especially in real-time, and monitoring, e.g., the volatile waste streams, transfer of nanomaterials via the air or toxicity in cell cultures, where measurement (in-)sensitivity can be disastrous for ecosystem and human well-being. The most prominent machine learning models contain the quantitative structure–activity relationships (QSARs) and nanostructure–activity relationships (QNAR) concerning the toxicological effect prediction of engineered nanomaterials using compartment-based mathematical models for toxicokinetic, toxicodynamic, in vitro and in vivo dosimetry, and environmental fate. EU reports, projects, and nanosafety clusters have assessed their applicability for regulatory purposes and to provide proper REACH guidance [21,45,46,47]. All these methods have been established for years, and the developed regulations/standards or SOPs satisfy the reproducibility and interoperability needs for the consolidated reporting of properties and behavior, while also satisfy the demand for safe nanomaterials by design. These methods have been used to map the profile of nanomaterials and their impact on both humans and the environment based on the characterization of morphology and physical properties [46,48], using the knowledge from fundamental research outcomes and mining new research and existing databases, which is facilitated when there is access to open data [49].

Another area to focus on is the storage of the produced process/characterization data, in order to maximize the impact of the experiments. Often, the volume of the generated data may seem large in materials science, but it may not concern computer scientists. Of higher concern is the ability to manage the extent of data and the useability across multiple domains [17]. A database plays a critical role, to digitalize the material development and automate tasks using modern AI tools. Un-/semi- and supervised learning includes the common techniques in solving classification and regression problems, while reinforcement learning algorithms demonstrate the utility to create programs by adjusting the model parameters, accuracy, and sensitivity via a “reward” system. This is often the ultimate goal, due to the fact that stored data demonstrate an incremental trend with time and the developed models should be constantly updated [10,31]. Another factor is that existing data usually have not been annotated and costly man-made annotation processes have to be followed in order to ensure that data are well-structured for the efficient establishment of predictive models [38]. Among datasets, different descriptors/features (rows or columns) are evidenced, and often one consideration is the “dimensionality curse”, which can lead to overtraining and induce bias in the developed models based on their observation frequency, and their ability to reveal hidden relationships in a family of materials is affected [12,27]. Some impressive demonstrations of machine learning include models to detect potential Heusler compounds and properties. Numerous descriptors have been utilized (22 descriptors) and performed predictions within over 400,000 classes in as short as 45 min [31].

So far, a lot of technical progress has been made in the field, as well as in documenting the recent progress and outcomes in the field. Below, a table summary (Table 1) is provided to present the landscape of reviews covering machine learning topics across different domains/fields to better position the scope of this review and the concepts covered in the following sections, which are connected to applications and implications for nanomanufacturing.

This review focuses on the need to establish digitalized mapping between materials descriptors that arise from different characterization/simulation methods and relate the chemical structure to nanomaterials options to favor the growth of dedicated properties profile and even to tune their production parameters to improve sustainability. In Section 2, the motivation is to introduce progress and challenges in discovering new materials by enabling shortcuts to rapid and safe-by-design routes, incorporate experimental parameters in order to enhance decision-making, and select the process parameters to produce nanomaterials to serve specific applications [45,49]. In addition, in Section 3, the improved capabilities for exploration of materials space and their corresponding digital representations are discussed and the key enabling steps to realize more flexibility in the production standards applied for selection of suitable nanomaterials, without the need to consider chemistry/physicochemistry or other complex theories in materials science, but using previous knowledge/experience documented in corresponding datasets [12,49]. Another important mission of machine intelligence is in the field of the development of safe and environmentally friendly nanomaterials. Hence, in Section 4, a threefold pattern in the prediction of their biological activity profile is presented based on data-driven representations, which could be prone to the availability of data and bias due to the applied procedures for sample preparation (contamination, purification), including biological systems containing human/algae cells, as well as in the bionanomaterial–cell interfaces. As presented in Section 5, the foundation for this scope is data mining, management, and curation to enrich new or existing databases, as it is commonly accepted that the catalyst for AI is the accessibility to structured and big data [59,60]. Consequently, the existing methods can be supported by the computational theory that focuses on data, which holds a lot of promise to greatly reduce the computational and experimental pressure for innovation. At the end of this review, the prospects of AI for the fields of nanotechnology, materials science, and nanoinformatics are summarized along with the conclusions of this study.

## 2. In Silico Materials Development

Nowadays, a key enabler in the coordination and management of the efforts towards materials development are the in-silico machine learning methods [3]. This includes the development stage, including the establishment of structure and property relationships, the simulation of the use-phase, as well as the prediction of the life- and cost-cycle for the raw materials use to improve the management of resources, the application lifespan, and the management of the wastes in the end-of-life. Knowledge of chemistry may also provide insights in surface modification of nanomaterials and their incorporation in reaction/transfer mechanisms for the reinforcement of interfaces in composite materials to serve advanced applications [23]. Concerning that the computational power is almost doubled every two years according to Moore’s law (expected to be valid till the end of 2025), it is accepted that machine learning has what it takes to take predictive modelling to the next level and support the decision-making process driven by data [3]. In the world of data, we know the needs that are intended to serve in different application fields, and nanostructures can be tailored in order to satisfy by design that the produced nanomaterial will comply with the high standards for commercial use of the produced material, while the impacts on health and the environment are also addressed during the production phase.

Machine learning efforts have been implemented on several occasions in order to accelerate the development of (bio-)materials in a bottom-up approach, such as in the case of peptides, and have been promising as enabling mechanisms to reveal and discover new therapeutics using deep learning to fight severe diseases such as cancer [39]. This multi-disciplinary field requires a lot of coordinated effort from scientists with different backgrounds in order to make sure that the predictive models are able to be validated by the theory and scientific principles and deliver what they promise, which requires the development of unbiased and scalable algorithms [3]. Especially, the computer programs should be optimized and provide an interface to ensure a user-friendly environment, the utilization of many different types of data, and to overcome dependencies on languages used, file formats, versions of packages, and enable the communication of real-world data with machine intelligence (Figure 2). The accuracy of the in-silico materials development can be improved by ensuring the quality of data [18], in close cooperation with the research scientists.

The main challenge of traditional trial and error materials development is the time-to-market which may take up to 10–20 years for their exploitation, while computational methods can efficiently reduce this timeframe to as short as 18 months [21,61]. The traditional material discovery method though, cannot adapt to the demand for large-scale and innovative fabrication of high-performance materials. A flagship project towards materials design is the USA “Material Genome Initiative” (MGI) was funded in 2011 [61,62]. A first systematic and mass effort on the collection of materials data was performed. “Material Genetic Engineering and Support Platform” is another project introduced by Vhina in the same direction, with the ambition to build a high-throughput computing platform. Several initiatives have since been established, which are dedicated to digitalization and automation in the material synthesis field, such as the Materials Platform for Data Science, Materials Project, Materials Data Facility, Novel Materials Discovery, Materials Cloud Archive, and other platforms for automated flow nanomaterial synthesis [3,10,21,23].

### 2.1. Sustainable Design, Engineering, and Discovery of Innovative Nanomaterials

In the past, the high-throughput screening of nanomaterials has been performed with molecular simulations or ab-initio calculations for the theoretical determination of their properties based on their chemical structure [63]. One current limitation is the efficiency when exploring the nanomaterials design space. The pluralism of nanomaterials and their corresponding functional properties could challenge the scientific community for years. Currently, there are many materials options which lack an application-oriented character, while any structured experimentation pathways have limited chances to lead to new knowledge and new nanomaterials invention. Even with the AI, design rules, and best candidates, the number of instances depends on the amount of data, while big data handling requires efficient algorithms since new obstacles can be introduced due to time and memory/processor demand.

More specifically, for the case of metal organic frameworks (MOFs), the design parameters and functionalization strategies should be optimized in order to identify synthetic paths for producing MOFs with an engineered distribution of active metallic sites within the porous network to deliver improved encapsulation efficiency and catalytic properties [64]. Predictive models due to their accuracy can efficiently assist the screening process considering that the hypothetic structural combinations are in the scale of millions, as well as the manufacturing parameters that can realistically enable their synthesis. However, deployment to all possible MOF structures is restricted in the case of molecular simulations, due to the high-computational cost. In the era of high-throughput nanomaterials design, machine learning has shown the potential to overcome such deficiencies with expensive calculations without a potential cost in the proper establishment of structure–property relationships. This is an enabling mechanism to aid the exploration of novel nanomaterials in unchartered chemical space and to identify the best candidate materials tailored to the application requirements [63]. The main bottleneck in this case, and for machine learning in general, is the access to enough and to high-fidelity data in order to enhance the innovation potential and identify hidden patterns, which contribute to the accuracy in future predictions. Since accessibility to high-quality data is satisfied, the impact of machine learning can be evidenced in critical areas, e.g., wise management of resources, time, and energy use, which contributes to the sustainability and viability of nanostructures development at the industrial level.

Machine learning has found success in nanoengineering, and more specifically, artificial neural networks (ANNs) were utilized by Li et al., in the modelling of pulse electrodeposition for composite Ni–TiN coatings to tune the nanogranular structure. The data were obtained from 45 different steel substrates to predict the sliding wear resistance of coatings with an error in the region of 4.2%. It was shown by the authors that a bigger grain size of TiN is evidenced when selecting a lower current density and a prolonged pulse interval, while the optimum wear resistance is obtained for the average crystallite sizes in the region between 39 and 58 nm [65]. In another study involving the physicochemical profile of TiO_2_ nanoparticles, the zeta potential prediction was enabled by ANNs. The temperature, pH, ionic strength, and mass content of aqueous dispersions were studied, with pH being the most influencing descriptor. Thus, the constructive model was regarded as a step beyond currently applied statistics, and more towards research on space exploration considering a wide range of conditions in silico, which due to technical and time constraints cannot be experimentally validated, but can offer a guidance on the next experiments. The impact in different industries, such as pigments and pharmaceuticals production, minerals processing, and construction, can be realized by tailorizing the synthesis or the manufacturing parameters towards an increased zeta potential value to favor the minimization of nanoparticle agglomeration events and to improve the sustainability of production when nanoparticles are used as reinforcements [66].

Of course, the utility of ANNs for structural design and process optimization has an imminent impact in the exploration of the design space. However, it is process specific. Such algorithms can be enabling as soon as they are shared, but what cannot be shared with success are the models developed by training on limited amounts of data. Overtraining, which is connected to the excellent fitting of known data, is limiting when the model is deployed to new data, and what is required to go beyond that is the ability to access thousands of data. In this way, the resources used for training a model can reduce the time investment for new users seeking to use an already developed model to explore amongst a rich design space since the model can be reused due to generalizability and speed criteria which will be met.

Graphene research is another bright engineering field due to the ability to tune multiple factors at nanoscale, such as defects, edges, vacancies, corners, dopants, reconstructions, and adsorbates, which determine the remarkable properties of graphene in the form of optical transparency, electronic, mechanical and thermal properties. In theory the perfect graphene does not enable to utilize the bandgap functionality in transistors, which is facilitated by the engineering of a bandgap up to 0.3 eV by introducing Stone–Wales and vacancy-type defects, or to control reactivity and induce local charge by controlling the density of π-electrons. Motevalli et al., in their study modelled graphene oxide template structures to increase the insights in the frequency of defects and broken bonds based on structural features, which significantly affect the strength and the conductive properties of this modern nanomaterial. The dataset contained 20,000 different electronic structures, which were obtained by simulation, and machine learning was used to establish a predictive Bayesian network model using 829 structural features, while 223 features out of them were proved sufficient to achieve the development of an accurate model. For instance, in the case of graphene oxide the size, shape, edges and corners can be correlated to the distribution of broken bonds. The distribution of hydrogen atoms was found to be representative of the lattice rupture, while the presence of oxygen functional groups is connected to actual number of broken bonds. Moreover, the presence of ether and hydrogen was shown to affect the integrity of the carbon lattice, which can be used to enhance the impact properties and overall performance. Their methodology was optimized by the incorporation of the density functional tight binding (DFTB) method on a training set consisting of 20,396 instances. The modelled graphene oxide templates represent a surface varying from 320 Å^2^ to 2457 Å^2^, several functional groups (epoxide, ether, hydroxyl) at various compositions, and morphologies (triangular, rectangular, rhombic, hexagonal). This analysis corresponds to over 8,500,000 possible adsorption sites, and the model demonstrated generalization ability for developing transfer learning functionality to other nanomaterials with known structure, which emphasized the significance of the utilization of machine learning models in engineering/design decisions of graphene structures, as well as the contribution to nanoinformatics [67].

In another case, machine learning has been used for the revelation of intrinsic electronic properties of graphene and rapid identification of the desired electronic properties by correlating the nanoscale features of graphene with or without undergoing functionalization treatment and represented by a structural footprint. The dataset is structured by 622 material options of computationally optimized graphene and the electron affinity (E_A_), energy of the Fermi level (E_F_), electronic band gap (E_G_), and ionization potential (E_I_) parameters have been modelled by Fernandez et al., demonstrating a high accuracy metric of *R*^2^ ~0.9 [62]. The benefit of this approach is the ease of acquiring the experiment structural properties of graphene by characterization (surface area, geometry, edge type, aspect ratio), while often the characterization of the electronic properties is challenging, especially when graphene is produced at scale. In this study, the relevant electronic properties were acquired using DFTB simulation, while graphene structures were consisted of 16 to 2176 carbon atoms. GA optimization was applied to choose the best hyperparameters and features combinations to access a universal solution of the developed model, until the fitness score remains unchanged for 90% of the generations. This in-silico, high speed screening approach can facilitate future research by reducing the experimental time and investment required to gain deep insights in graphene nanostructures, especially when there is access to virtual libraries containing experimental data.

Machine learning has been also deployed for the precise prediction of crystallographic orientations at nanoscale to assist the experimental design and the lithographic preparation of those structures. In this case, Fernandez et al., used the electronegativity values of graphene nanoflakes with an absolute error lower than 0.5 eV and their molecular graph information to facilitate the rapid prediction of the energy gap and the determination of the topology of the nanoflakes accordingly. Such models can be used to moderate the control of molecular connectivity and edge characteristics and can find application beyond graphene structures with other 2D nanomaterials, and effectively support the screening process, as well as defining the correlation of functional and structural properties, especially when there is access to libraries with relevant data. In this study, the most accurate predictive model was established with support vector machines (SVM), optimized using a genetic algorithm (GA) in the background procedure, with a topological resolution distance of 1–42 atoms [44].

The present status of graphene research and machine learning is currently guided by the simulation data, which can provide fundamental insights in the structure property relations offering larger training datasets which can feed the data-driven models and reveal properties relation which could not be traceable by human alone. The main concern is that the simulation methods generate data in a structured way while real materials production can introduce a manufacturing footprint, including structural defects and presence of randomly distributed heteroatoms affecting their properties. This is currently limited by the throughput, scale, and limitations of current characterization techniques which cannot provide the experimental validation of the simulated structures. Thus, it is challenging to actually connect graphene structure and manufacturing considering also the fact of the statistical character of graphene powders and films produced, which currently are sampled to extract any structure and property information. However, still, the mapping of intrinsic features of nanomaterials generated in-silico can provide a solid basis to push innovation and sustainability by identifying the best candidates for nanomaterials application and improving the resources allocation of goal-oriented experiments.

In-silico models for carbon nanodots (CDs) were developed by Han et al., to support the parameter optimization, which is a challenging task for these sophisticated materials which serve special applications due to their exotic properties, such as Fe^3+^ ions sensing in solutions. In this case, the research bottleneck is identified in the quality of the process parameters, which often suffer from noise and their wide exploration space, which cannot be handled by a single researcher. Machine learning can address this challenge with success and assist the screening of high-quality CDs, due to its effective prediction, optimization, and fast acquisition of results capability. This study focused on the hydrothermal synthesis, which is well-established, and the prediction of the process-related fluorescent quantum yield (QY) was based on several descriptors, such as EDA volume, mass of precursor, reaction temperature, temperature ramp, and reaction time out of 391 experiments. More specifically, the XGBoost-R algorithm guided the experimentally verified synthesis of CDs out of 702,702 available combinations, exhibiting a strong green emission with QY up to 39.3%. The more important features for the engineering of CDs were shown to be the mass of the precursors and the volume of the alkaline catalysts in order to achieve high-QY and successfully bridge the gap between theory with the experiment, while even chemical descriptors incorporation can further support advanced research in the future and generalize the machine learning model [68]. By providing the framework and sharing the methods and algorithms, there is the potential to cover the engineering aspects of the aforementioned graphene development approaches.

In another case, nanomaterials were in silico designed by Dewulf et al., to provide feedback to the bioinspired in vitro synthesis of high porous silica as a green alternative to the conventional synthesis. The bioinspired synthesis is characterized by the short reaction duration, the mild conditions at room temperature, and the use of an eco-friendly precursor (sodium metasilicate pentahydrate), which satisfy a viable and scalable approach able to offer promising deployment at industrial scale. In this case, the best-case scenario is that an agreement of predictive modelling and experiment occurs. Practically, what is needed is to provide a machine learning basis that beyond predictions can follow the experimental sequencing, apply automated corrections and crossvalidations to improve the model, and propose new experiments.

Furthermore, the sustainability of the process can be further supported by such a design of experiments by performing global sensitivity analysis in order to support the resource-efficient product and process development. In their approach, they used a factorial materials design 2^3^ to improve reaction yield and surface area, which included the parameters of precursor concentration, pH, and reactant concentrations in the form of silicon to nitrogen ratio, in combination with optimization using a central composite design, leading to a yield of 90 mol%, while the highest surface area value that was obtained was 400 m^2^ g^−1^. Since the aforementioned properties are critical factors to ensure the successful commercialization of the materials, regression analysis was implemented along with global sensitivity analysis using the Sobol’ index in order to further improve the process. A central composite design and multivariate analysis were used to model the experimentally determined outcomes to assist the rapid identification of interactions and parameters that are correlated with physicochemical properties with high precision, using a wide parameter and experimental space. The main parameters involved in this machine learning approach were Si precursor concentration, pH, and the Si to N ratio to predict the reaction yield and the surface area. It was shown that Si precursor concentration and Si:N ratio determine the precipitation occurrence. An optimization regarding the reduction of the effort spent in experimental verification was realized by using a sequential design in order to efficiently perform pre-screening and screening, and subsequently the optimization of the experiments [69].

Finally, in silico materials development has been demonstrated for organic structure directing agents (OSDAs), where eight different models were utilized using evolutionary algorithms. The dataset was consisted of 1,000,000 trial molecules, which were generated by MD. Machine learning was used for the prediction of the stabilization energy in comparison with the respective output of MD in order to decide on the synthetic pathway. The actual number of the compounds with a stabilization energy below −15 kJ/(mol Si) was conducted using ANN supported by an optimization generic algorithm, which resulted in a lower number of molecules. The training dataset was consisted of stabilization energies for 4781 which are going to be developed on putative zeolite, which were obtained through computationally intensive MD calculation, resulting to the in-silico generation of 469 exceptionally stable structures. Thus, an effective strategy for the design of OSDAs was proposed for zeolite beta with high correlation to the MD results [70].

In order to go beyond the in-silico materials design and design space exploration, machine learning should utilize real world experimental data, satisfying the data quality standards. Another challenge is connected to the speed of experimental data acquisition and the amount of data analysis by each measurement which is dependent on human resources. Machine learning advances can efficiently automate the extraction of materials properties, e.g., in the case of carbon nanotubes (CNTs) to measure the diameter of a greater amount of instances compared to a single user, and for effectively increasing the statistical sample, and thus robustness of the derived data.

### 2.2. Contribution of High-Resolution Characterization Coupling with Machine Learning and Computer Vision to Structure High-Quality Materials Datasets for Materials Development

Machine learning comes to bridge the gap of high-quality materials information acquisition form high resolution techniques at high throughput by utilizing methods to support and detect the target features and properties that can be used for the in-silico development of materials or for the prediction of unknown properties of new or existing materials. Computer vision is also a tool that should not be excluded, noting that image analysis could deliberate automated processing of the recognition of key mechanical properties (stress-strain plots), and extraction of the mechanical properties of produced materials similar to a human analyst. The most prominent machine learning libraries for computer vision are Torch, Theano, Caffe, TensorFlow, and Keras [17]. Already, scanning electron microscopy (SEM) has been used for identifying the successful synthesis of nanomaterials, to evaluate material quality by the observation of its surface, to investigate the (de-)bonding of hybrid structures, as well as the shape and distribution of the dimensions, and several efforts have been made to develop image vision AI models to support the automatic classification and annotation of images from different materials built on existing databases (over 150,000 images) with accuracy almost 90% [38]. 

Sophisticated focused ion beam (FIB) characterization also uses digital image correlation for obtaining insights about the stress gradients and for imaging soft materials such as filled reinforced polymer nanocomposites [71]. AI coupling with this technique has led to effective image-processing to achieve a super-resolution of 3D images and to reduce the observation times, by demonstrating superior restoration as the asymmetric resolution is increased. Moreover, images from X-ray spectroscopy surface mapping (EDX, EBDS) or Raman mapping are utilized for further image analysis, e.g., in clustering to identify the materials microstructural composition, which could be relevant to martensitic transformation and phase nucleation monitoring for steels thermal processing via elemental maps, for the examination of the bonding state in composites and hybridization of the bonds in the interfaces.

Another interesting high resolution imaging technique, scanning tunnelling microscopy (STM) [72], was coupled with deep learning, which was utilized for automatic particle recognition, while the model “ParticlesNN” was deployed online as an open source tool to facilitate the extraction of nanoparticle information. The main benefits provided are the ability to handle images containing noisy data, to perform statistical processing in the form of histogram and tables for all the identified nanoparticles. The “ParticlesNN” web service also provides the flexibility to classify particles in the micrometer scale (with lower resolution limit than the technique used for establishing the machine learning models), while the input images can be derived from different instruments, such as SEM, due to the similarity of the image output. Thus, it is possible to maximize the output of an imaging technique and support the increased accuracy, quality of the material descriptors, and the number of annotated instances, to feed the machine learning models and establish a statistically representative connection between structure and properties.

Machine learning was used by Lee et al., in order to overcome the characterization techniques challenge of sensitive and accurate characterization at nanoscale. Single particle inductively coupled plasma mass spectrometry (spICP-MS) is a prolific method in this field and outperforms other conventional techniques used, such as dynamic light scattering (DLS), which was used to acquire data on Au nanoparticles size distribution and measurement of their concentration with the highest possible precision. K-means clustering was used to process the raw data for the improved discrimination of the signal, removing the background noise and quantitatively resolve different size groups with a resolution lower than 2% by mass and 20 nm by nanoparticle [73].

Current efforts are limited though in the information extraction from nanoparticles with circular crosssections and spherical shapes. Machine learning can unlock maximum potential when deployed on imaging and other shape-specific characterization techniques when more geometries can be identified and conditional functions assess critical materials structure features, i.e., inner and outer diameter, length, curvature, deviation from circular shape, etc. Moreover, current analysis is often limited to 2D information, which in case spherical particles can be extrapolated to 3D information. However, 3D information is often needed to describe nanomaterials with complex geometries.

Horwath et al., used computer vision and ETEM images to improve the image segmentation capabilities in an automated way and deploy a model suitable for nanomaterial detection which is limited only by the resolution of the characterization technique and not by the machine learning model ability to identify and localize features. In their approach to this common segmentation task, deep learning regularization of the input datasets was shown to be the key factor to establish an effective model rather than developing an architecture with convolutional neural networks (CNN). This is connected to the selection of boundary pixels, which requires the sematic information distributed in the image, which may be lost by increasing the variance of the intensity histograms [74]. The knowledge of data features, as well as hypothesis-oriented design of the predictive model, were shown to address common challenges in materials science informatics, while class imbalance, overfitting, and accessibility to sufficient amounts of data limit the prediction efficiency. The proposed architecture is quite simple, consisting of a single convolutional layer, which satisfied an efficient computational performance in regard to accuracy and the ability to quantify results, while the model kept the prediction standards close to the state-of-the-art (SoA), by adopting suitable metrics that evaluate the limitations induced by several descriptors. Interoperability of the developed model is also satisfied by the fact that computer vision models are not limited by the acquisition characterization technique, but by the instrument resolution limits.

Similarly, Ilett et al., introduced the ilastik tool for object classification, which can be tailored to a wide range of parameters that can be used to deconvolute particles by microscopy images [75]. This tool includes a function of detection of different particles even in agglomeration state and corresponding quantification features which could be used to monitor nanomaterials stability and obtain more insights in the interpotential dynamics and the tendency to agglomerate, which is a critical aspect in nanomanufacturing. Thus, machine learning was introduced to provide in that case the capacity to overcome the time-consuming manual process for identifying the agglomeration tendency and projected shape of agglomerates, as well as to overcome the limitations of DLS characterization, which as discussed above is prone to missing the agglomeration events since a circular shape of nanomaterials is assumed in all cases. In addition, it has been argued that DLS information on colloidal stability can be limited since the suspensions studied with this method are stable colloids.

In summary, with the exploitation of high-resolution computer vision and other machine learning models, it is possible to detect the production footprint in regards to the (nano-)materials properties and provide the actual characterization mapping over a wide range of instances. By using previous knowledge in terms of characterization output and images it is possible to automatically extract high number of materials parameters and high-quality data, which can be used for training to extrapolate the prediction to multiple material features.

Several approaches include randomization of the parameters of the process to achieve the suitable material properties, which requires access to a large variety and unbiased data. Usually, particle swarm optimization (PSO) and GA have been shown to efficiently support the design of the experiments and support the optimization tasks using the minimum number of experiments with this data-driven approach. Thus, machine learning can efficiently help scientists to obtain high-throughput information from characterization techniques and to utilize previous raw research or metadata to reveal unforeseen physical/chemical properties, topologies, and stability (agglomeration tendency), high quality features, and the inner structure of materials with high precision [12], which enhances the innovation potential.

### 2.3. Optimizing Formulations and Composition in Nanocomposite Materials Engineering and Additive Manufacturing to Improve Performance and Support Applications

Additive manufacturing (AM) has raised a lot of attention in recent years due to its prominent technological role in shaping and (nano-)manufacturing complex metal-, polymer-based, and nano-reinforced components to serve demanding engineering applications with the unmatched benefits of providing zero waste “bottom-up” scalable manufacturing solutions for complex architectures. AM can be realized in desktop microfactories with many benefits, such as reduced needs for additional tools to be used, functional parts can be delivered without the need to assembly, and with minimum requirement of down time, products can be customized and meet the societal needs [76,77,78,79,80]. The role of nanotechnology in this field lies in customizing and engineering the material and nanocomposite properties by the utilization of nanoreinforcements, such as graphene and nanofibers, with high reactivity, surface area, conductivity, sensing capabilities, and modified surface chemistry, which can introduce multiple functionalities at the multiscale level [77].

One main research priority remains to replicate the properties of conventionally used materials with more sustainable material options. Current established research has shown how carbon nanomaterials (graphene, graphene oxide, CNTs, carbon black nanoparticles) can be used to formulate AM nanocomposites with improved tensile properties, thermal stability, conductivity, and radiofrequency-induced heating capability, and to broaden the application field of the developed materials. Moreover, it is strongly regarded by the community of AM that the vast amounts of domestic waste could be managed by efficient and sustainable upcycling strategies to lead the manufacturing of products with added value by incorporating nanotechnology to the feedstock formulations [81]. For instance, polymer nanocomposites have the potential to support adequate performance in many fields of application with specific needs for wettability, elasticity, durability, and conductance. AM process may be hindered though, due to the rheological behavior/clogging/homogeneity of mixing at high mixing ratios of nanomaterials, which may be required to reach, e.g., a workable conductivity, stiffness, etc. [82].

AM digital character can lead the digital nanomanufacturing era empowered by AI. The AM workflow involves the materials/object virtual and reverse design (layers structure, composition, nano-fillers, architecture, aesthetics) from the smallest volume of reference in the form of droplets, powder, wire to fabricate flexible and lightweight components. The AM process digitalized character offers the opportunity to fully digitally controlled operations that can be advanced and accelerated with the use of smart, high-precision, data-driven tools and metrology to introduce online and real-time process and on-demand adaptation capabilities, and tackle current variation in product quality, thus increasing confidence and reducing unpredictability concerns [50,78,79].

In order to bridge the gap of current manufacturing technologies and smart factories, AM infrastructures should enable to go beyond the current SoA; (i) regarding the dependence solely on feedstock screening operations for process assessment, and (ii) the “open-loop” system operation and introduction of sensor systems for online feedback measurement to enable smart process self-adaptation (control, quality assessment, calibration, monitoring) supported by AI. In this direction, Banadaki et al., proposed correlations feedstock-sensor data-process parameters-characterization data by establishing a schema for interactive and scalable machine learning development to support the reliability of the process in a cost-effective manner, and to improve the end-products from AM [83]. In this scope, it is essential to incorporate (open and interoperable) ontologies in order to facilitate the knowledge management of AM digitalized and structured data and support the AI to reveal and discover reasonable and currently unseen/new knowledge [84]. Already relevant progress has been realized by Granta additive manufacturing, Senvol Database, and Senvol ML, which is expected to evolve and flourish in the next decade.

Amongst the applications of AM, the ambition and the challenges that are expected to be confronted in the next years are related to bio-based applications from the fabrication of drug delivery devices/sensors to the manufacturing of polymer-based synthetic tissues and organs reinforced with nanomaterials. A strategic benefit in this direction is the ability of AM techniques to combine precursors with different properties to represent the human physiology, which often requires hard and soft segments (both cellular and acellular) assembly to sustain the stresses induced in human body environment. Several techniques have been adopted in this field, including selective laser sintering, stereolithography, fused deposition modelling, and bioprinting (extrusion-assisted, laser, droplet, ink) [77]. Engineered bio-compatible materials for slow-release medicines and tissue engineering can perform occasionally even better than natural materials, and thus AM can have a great impact in the field by providing easily accessible, low cost, and faster address the market needs for health care technologies across the globe [81]. The next generation in miniaturized sensor devices has been realized in the AM industry, with recent trends in developing a lab-on-a-chip (LOC), to track (bio)chemical processes in the clinical diagnostics sector and perform fast and easy assessment. Other benefits of this technology include the reconfigurability, modularity, portability, compactness, low power consumption and electronic noise, embedded computing capability, and the highly localized topology, which promises the analysis of a specified point of care with improved sensitivity and minimum resources. Engineering the LOC devices enables the control of microfluidics, which is key for the electrokinetic or micropumping control of fluidic transportation and efficient separation when examining liquid samples with high precision in several conditions; when the flow is continuous, or droplet-wise sampled. Thus, this is an important technology for biomedicine advancement following the automatic continuous tracking, which can be used for online feedback and adjustment of multi-material AM processes and provide tailor-made microfluidic micro-electromechanical systems (MEMS) [85].

In a different bio-based application, Zafeiris et al., used AM and 3D CAD representations to fabricate hydroxyapatite/chitosan scaffolds with controlled porous network to facilitate the cell attachment and cultivation requirements to enable the functionality of tissue growth and the proper transfer capability for nutrients to reach the cell cultures. The direct ink writing method was used to enable the realization of the regeneration of bone tissue and successfully develop scaffolds mimicking a proper extracellular matrix [76]. Another study case of biomimetism with the use of machine learning in AM contains rapid design solutions among a vast design space in the field of biomimetic design. As an input simulated metamaterials and stored data have been used by Gu et al., in order to develop a self-learning algorithm to identify the best candidates for the production of high-performance hierarchical materials with highly defined microstructural patterns. Compared to conventionally used FEM for the prediction of mechanical properties it was shown that machine learning can induce a shortcut to long computational requirements from 5 days to less than 10 h and up to 30 s for training of the algorithms, which are able to screen at high-throughput (billions of designs per hour) [86].

In addition, and beyond healthcare industries, AM can serve a range of applications in automotive, energy, aerospace, due to its unique engineering features, with one main identified bottleneck regarding the surface integrity. An intelligent and digitally controlled methodology was developed by Li et al., who introduced a sensor-based (accelerometers, infrared temperature, thermocouples), data-driven ensemble model for the surface roughness prediction [87]. With regards to the wider industrial exploitation of AM, product quality assurance, processing defects, access to materials libraries, and design for AM remain limiting factors. Machine Learning as standalone or combined with physics modelling can be the catalyst in this direction, especially by the scope of discovering/generating and predicting the performance of AM metamaterials, namely, elastic/shear modulus and Poisson’s ratio in an automated manner based on the desired properties (virtual experiments). What is more, machine learning can assist the on-demand and reverse process adaptation to improve quality control during manufacturing, assist the precise control of printing topology, including melt-pool geometry for DED processes, and enhance the feedstock screening at a pre-manufacturing planning level. Another important quality parameter of the engineered AM products concerns the mesoscale porosity, which is process dependent and highly related to the mechanical performance. A nullified porosity is the ultimate objective in metal AM to achieve full dense structures in order to obtain adequate fatigue properties, while in biological or energy absorption applications a controlled porous architecture is required. A machine learning paradigm for the latter case is to correlate porosity with manufacturing parameters using neural networks (NN) or an adaptive-network-based fuzzy inference system [51].

The role of machine intelligence has risen to support the advancement of AM processes and materials evolution dedicated also to different sectors. By using data-driven predictive modelling, it has been possible to identify and correlate the microstructure of materials to thermal stresses for metal components. CNNs have been successfully utilized to establish such structure–property relations. Bhutada et al., demonstrated the identification capability amongst six different microstructures by utilizing feature extraction via k-means clustering on images and subsequently image vision models were established to classify each microstructure with over 97% accuracy. The correlation with the principal, hydrostatic, and other stress tensors was conducted via regression among the six microstructures, which enabled quantitative comparisons of internal stresses of model-based predictions and experimental values with high confidence. Another outcome regarded the NN model representation as a reduced order microstructure, which can be used in conjunction with FEM to successfully predict thermal stress on an 3D printed components [88].

AI has also found application in online and real-time video monitoring (single shot detector) of the fused filament fabrication (FFF) process. The detection of stringing defects generated during printing was enabled by the development of deep convolutional NN, which established the connection of the defect patterns with the printing parameters without a need for machine or camera calibration, in order to enable the fast online adaptation of the process. Successful case studies in the powder spreading stage via computer vision have been published also on the automated classification of unwanted phenomena and anomalies (splatter defects, delamination) for the live monitoring of power bed process [50,89]. Parameters, such as the size of the printed device, distance to the print area, and camera resolution, have an effect on the precision of the computer vision, and by using AI it is possible to adapt the parameters in order to eliminate print errors [89].

Laser-based AM is another (nano-)manufacturing method that is competitive due to the capacity of rapid prototyping at reduced cost and the ability to work on flexible and curved substrates, providing the advantage of forming wearable electronics with the involvement of nanotechnology. Especially for metal AM, nanomaterials can tackle the challenge of the feedstock supply and quality requirements regarding the size distribution and uniformity, but size and shape still affect the success of nanomanufacturing [90]. Real-time and intelligent control of the AM process is essential in this scope to minimize the defects created during printing and save energy by mitigating the need for post-annealing treatments after fabrication, while the throughput is increased [50,90]. Another case is the utilization of a focused laser beam for the optical printing of nanomaterials dispersed in proper solvents on different surfaces [90].

Self-organizing maps (SOMs) represent another machine learning method which has been used for assessing the AM profiling accuracy compared to digital CAD models to establish relations with the parameters of processing such as extruder temperature and infill percentage that may lead to deviations. Such a causal relationship establishment was outshined by the authors as an imminent measure to efficiently adapt the process and improve the quality. The universality of this method is connected to the commonality of the features correlated, which can bring intelligence and new design rules to all AM methods that confront such qualitative challenges [91].

It is clear that machine learning holds a lot of promise for the establishment of data-driven supervising tools in a closed-loop AM process. Especially, SoA powder bed printers lead to variation in the mechanical properties from experiment to experiment. Automation in defect recognition via image analysis with NN has been outlined by Razaviarab et al., as a gamechanger in smart AM automated adjustment of processing parameters, which is expected to limit the wastes produced and reduce the effort and energy wasted [50,92].

The proceedings in the field of AM seem to be encouraging regarding the adoption of attractive and intelligent tools in the production lines. The main concern remains concerning the willingness of AM machine manufacturers and software developers to open channels and enable deep adaptations up to software/firmware since intelligent algorithms have to be incorporated directly to the digital ecosystem to enable process adaptation and real-time control, as well as product optimization.

As it has been shown already, ANNs dominate in a considerable amount of publications due to their unique ability to solve complex real-life engineering problems beyond AM, by the utilization of previously measured characterization data and due to the ability to achieve significant time savings [41]. The interpretability of ANNs in decision making has been popular in data science, commonly by solving the inverse problem, where the experimental data are correlated to the microstructure and functionalities of nanomaterials to gain new insights by using first-principles-based tools to accelerate the time-to-market of novel nanomaterials/nanocomposite materials/systems [93]. The attractiveness of this method is in its simplicity based on the previous experience based on characterization results, without the consideration of a sound theory, which cannot be substituted by machine learning, but used complementarily [94]. For instance, an ANN predictive model was developed to predict the pool boiling heat transfer coefficient (HTC) and design refrigerant-based nanofluids containing CNTs, TιO_2_, nanodiamond, and Cu with correlation coefficient (*R*^2^) of 0.9948 and overall mean square error of 0.0153. In this case, data were mined from literature papers resulting in a dataset unfolding 1342 different experiments, and seven descriptors were included, namely heat flux, saturation pressure, base fluid thermal conductivity, nanoparticle thermal conductivity, nanoparticle concentration, lubricant concentration, and nanoparticle size. The pool boiling HTC of refrigerant-based nanofluids was determined over wide ranges of operating conditions and the best functionality was demonstrated by using a simple one hidden layer architecture with 19 neurons, while tansig (hidden layer) and purelin (output layer) were used as transfer functions [95].

In another study, Demirbay et al., used a multi-layered feed-forward neural network (FFNN) to predict electrical conductivity in polyesterene (PS) doped film coatings reinforced with multi-walled CNTs (MWCNT) to improve electrical conductivity. The dataset describing the formulation features contained the concentration of surfactant, initiator, MWCNT, molecular weight, particle sizes of PS latex. Training regulation was performed using a Bayesian backpropagation algorithm. In this case, the ideal topology of the FFNN was evaluated using several metrics, such as mean squared error (MSE) and the determination of coefficient (*R^2^*). The optimal architecture of the network was consisted of eight nodes in the hidden layer using a log-sigmoid transfer function for the training, which was confirmed by the *R^2^* value in the training phase, and with the MSE value in both training and testing. The relative importance-based sensitivity analysis also showed that the concentration of MWCNTs influenced to a greater extent the conductivity results. Finally, a mathematic model was used to introduce training weights and reduce bias in the predicted results, which were in agreement with the experimental values [41]. Therefore, an explicit mathematical function was established for the prediction of the conductivity using weights and bias values.

Ashrafi et al., in their study of concrete nanoformulations used a feed-forward back-propagation network with a specific architecture containing 22 nodes and one hidden layer to acquire the force–deflection curve and the 28-day compressive strength. The results were in accordance with the experimental output, while two additional methods were proven to provide even better prediction efficiency, namely standard deviation normalization and the Levenberg–Marquardt algorithm [96].

In the study of Huang et al., the interesting effects of CNTs reinforcement of cement composites were investigated. More specifically, flexural and compressive strength were predicted using ANN and support vector machine (SVM) models, which were trained and tested on literature data in a sample of 114 experiments. Several aspects of modelling, such as the size, the quality of the dataset, experimental factors, and undefined parameters, strongly influence the mechanical properties. All parameters seem to lead in deviations of the predicted and actual values. Among the parameters used in this investigation the authors revealed that the length of CNTs has the highest impact on the output values of the compressive strength. Regarding the flexural strength, curing temperature was shown to be the most influential parameter. Both outcomes support the formulation design and the selection of nanomaterials to achieve the desirable optimization of the mechanical properties. More insights about manufacturing, namely sonication energy and time, additive, curation humidity, and other experimental descriptors, may enhance the predictive ability of the established model [97].

Another demonstration of ANN capability was used to utilize 75 different material coefficients to model the microhardness of metal-ceramic nanocomposite coatings in order to select the proper combination of non-investigated experimental parameters (factor model—limited by the variation range of the coefficients) and deliver the desired properties with high accuracy. This model contributed to the data-driven optimization of the nanogranular structure of a wide variety of nanocomposites, including FeCoZr–Al_2_O_3_, Fe–Al_2_O_3_, Co–Al_2_O_3_, Ca–SiO_2_, Co–CaF_2_, Fe–SiO_2_, and Co–MgO_2_ [94].

The main argument in the establishment of structure–property relations with NN architectures is that the number of layers, number of nodes, and the type of the network (recurrent, convolutional, feed-forward, etc.) may not be generalizable. For instance, when studying the mechanical properties of nanoreinforced composites with CNTs, titanium nanotubes, or Ag wires, factors, such as the different surface modification, length, weight concentration, catalysts used in the production phase, diameter, surface energy, mechanical properties, and conductivity, may require a larger number of hidden layers and more complex architectures to legitimate the relation of nanomaterial used to the composite performance or target property. Currently, in the framework of scientific research and publications, this is unsustainable as it requires severe investment and resources to perform all experimental procedures in regard to synthesis and testing. The research community can assist the needs for more systematic and structured information by sharing their data to open databases and facilitate the development of usable models.

Currently, it seems that data sharing is the resource that is actually reusable, rather than the actual machine learning model and this is problematic due to the fact the resources and computational energy spend in training cannot be reused, which also has an environmental impact. The inverse condition was thought to be the norm since the main discussion has concerned the generalizability and universality of models; the open models are as generalized and as fast as possible, providing a framework of assessment tools for knowledge exploration, as well as enhanced and decentralized decision-making. However, cooperation and sharing are required to realize actual progress in this direction.

## 3. Optimization of Materials Synthesis Using High Throughput Screening Evolutionary Algorithms—Reverse Engineering

A special field of process design is reverse engineering, which will have a key role in the Digital era of Industry 4.0 and the next generation of human-oriented transition of Industry 5.0. Materials are defined by their properties and their performance under operating conditions, which are highly dependent on the manufacturing parameters. Rational material design is an ultimate goal in the fields of modern materials science and engineering. Finding the ideal conditions to tailor materials properties, or for reproducing a mimetic design (often it is the case for bioinspired computation) for candidate/substituting/greener materials, is expected to overtake the market. Knowledge and experience gained on industrial materials development can feed with special features machine learning algorithms and demonstrate the functionality to search beyond the knowledge space.

Currently, this is a demanding process, and effort is spent amongst these communities to compile extensive datasets of materials properties in order to improve accessibility to resources containing materials properties, which are key to establish design rules and intelligent tools. Previously developed rules or models demonstrated a semi-empirical character and were based on human intuition and knowledge, while machine learning can take materials design to the next level by combining automation, accuracy, and rapid calculations. Digitalization of materials and processes is expected to support a variety of new applications, while actual advances in data analytics and routine methods are expected to maximize the impact of raw data to identify quantitative descriptors for nanomaterials to support the optimization problem [21].

What is expected to make the difference with machine learning is the ability to take the most out of non-crisp information. Experimental information, especially at laboratory level contain most of times granular information, such as temperature and the convention for performing the experiment at laboratory temperature (could vary from summer to winter season and during daytime), which is usually not as precise or reproducible in regards to the strict requirements set by the scientific community (fuzzy-probability-granular information). This information is often not regarded in the traditional decision-making process, as it would limit fairness and accuracy of the approach [98]. Thus, machine learning and predictive modelling can give a competitive benefit on this case by taking advantage over granular information, which is inevitable in several cases, and successfully support a realistic representation of any specific problem.

The optimization in materials synthesis often is a challenging task, and machine learning models should be supported by optimization algorithms to settle on a generalized representation of the material and solve the task with a universal solution. In this scope Figure 3 summarized the end-to-end nanomaterial development aspects. GAs are commonly used to serve the optimization needs and resemble the Darwin’s theory; in this case the more accurate model survives to the next generation. For instance, innovation in the field of chemical compounds utilizes the known compounds by attributing a mutation factor. In the next generation, the novel compounds are evaluated for their properties, and since a weakness is identified oriented by the application field and user defined rules (output of the model), the compound is disregarded and the rest is used to produce the next generation, which inherits part of their properties. Thus, it is possible to achieve shortcuts in experimentation, while key properties, such as conductivity or hardness, can be optimized [17].

Another functionality of machine learning regards the possibilities to adapt to limited access of data and address optimization concerns, which may be the case in academic research, where often there is limited access to data concerning the manufacturing and characterization of materials. Moreover, experimental applications in materials optimization typically corresponds to small batches in the scale of mg–g, which is understood to limit the process scalability, due to the constraint that mass transfer parameters cannot be deployed to larger volumes of production [61]. Model averaging and bootstrapping have been two prominent methodologies to handle problems where only a small dataset is available, which is a data-oriented limiting factor regarding model development, which leads to simplified models [30]. Model averaging enables the building of powerful algorithms by the combination of different simple models, which can be in the form of a ‘committee’ or an ‘ensemble’ model. Ensemble-based algorithms are extremely flexible, less prone to over-fitting and outlier sensitivity, and are more amenable to tuning, but are more computationally demanding than ridge regressors [26,30].

The predictions of model averaging aim to combine models with predictive efficiency beyond random chance, leading to a more objective model, which is mostly implemented through ‘bagging’ in the case of decision trees random forest algorithms, which can reduce the model sensitivity to the noise of data during the training phase. Another method used in literature is Bayesian model averaging, which is mostly applicable to linear models by using a weighted average of each prediction based on accuracy. ‘Stacking’ of models is another methodology to combine models, where each prediction is fed to another ‘second-level’ model, which utilizes the best capabilities of each model used for stacking, which has been shown to well mitigate model-specific overfitting [45]. In this way, machine intelligence can overcome model overfitting concerns and generalize, combining the flexibility of adding a large number of models and avoiding ‘missing out’ on the predictive capabilities of each individual algorithm [32]. For example, Barnard et al., in their study utilized a dataset consisted of 5 features about the shape of gold nanoparticles (GNPs). 4000 different synthetic routes were used to optimize the development of these nanomaterials, and 70 different models were developed [30]. In this case, strong correlation of the features resulted in an overfitting issue, which can be handled by leaving out the correlated feature, or by any of the abovementioned averaging/bootstrapping/stacking procedures.

However, one main bottleneck commonly confronted is related to feature extraction. Selecting the appropriate descriptors by implementing a Pearson parametric correlation map ensuring that informed predictions can be objective and trusted, towards the development of a model with unbiased establishment of parameters relation [99]. This is often a good strategy to avoid overfitting issues when developing machine learning models and improve the prediction accuracy, while excluding strongly correlated features [36]. Besides, bottlenecks related to computational resources and availability of data/descriptors can be overcome based on the parametric sensitivity analysis and dependence plots indicating the parameters that are more influential in the predictions, thus selecting the most suitable descriptors for establishing multi-perspective predictive models [100,101].

Till now, numerous success stories that use machine learning in materials science and nanoinformatics have been published. The applications of chemical discovery, though, demonstrate much room for improvement, and more specifically addressing theoretical and practical challenges can lead to a revolution in chemistry. The need for creative interdisciplinary approaches is highlighted to cover increasingly broader domains of nano- and cheminformatics, such as combining statistical and quantum mechanics, chemical knowledge, and prolific machine learning tools, which promise to advance the level of understanding in the field, as well as reduce the time-investment required in conventionally used simulation methods. Currently, there is a need to model and access information about electronic and energetic properties at nanoscale, while establishing proper (physicochemical) structure–property relationships. This is often limited to structural and configurational features. Another gap is the need for machine learning to learn how long-range electrostatic, van der Waals, and polarization interactions occur in a studied material. Currently, the description of local chemical bonding has been successful. The potential coupling of intermolecular interaction theories is expected to make realistic the better understanding of bonding and interaction states in more complex systems; considering the current level of maturity in machine learning there is an uncertainty level that should be evaluated and systematically overcome. Another consideration is binding machine learning with Hamiltonians to better describe the electronic state of materials, using the basis of density functional theory (DFT), and the methods referred to as “molecular orbital”, “tight-binding”, and “many body dispersion” techniques. Still, there is the computational efficiency bottleneck to overcome in this field of simulations [54].

### 3.1. High-Throughput Screening and Optimization of Nanomaterials with Genetic and Other Evolutionary Algorithms

Inverse design aims to use information of the desired materials functionalities and to use predictive modelling, with the aim of enabling the description of the chemical space through the establishment of proper structure–property relations. The traditional forward development depends on the evaluation of each experiment performed, while inverse engineering has a more goal-oriented character, which may be the key to guarantee success in fewer steps. In theory, the major problem with inverse design is the number of the possible configurations that led to the optimum nanomaterial structure, which corresponds to a number of candidates that may be as big as in the scale of thousand structures, which is limiting the experimental verification capability [21]. Another concern is that the predicted material structures may lie between the chemical space variation introduced by the features used for the training, and the selected features that are used to establish a predictive model [45]. In this direction, evolutionary algorithms may be the answer to this challenge. Genetic evolutionary algorithms beyond their use as single algorithms can be also combined with back propagation ANNs “GENOUD” algorithm, which was used by Liu et al., to utilize the global searching power of the GAs to tackle the main problem of back propagation algorithms; by introducing training weights it was managed to avoid local optima as the predictive outcome solution of a specific case study. Moreover, the convergence speed is another advantage of the GENOUD algorithm, due to its nature (in the family of traditional gradient based optimization algorithms), which can improve also the speed of ANNs [102]. As it is shown, machine learning can be prolific in the field of high-throughput screening, which has been already demonstrated with success by researchers [21].

What is more, machine learning can lead the establishment of effective strategies to utilize the outcome of failed or partially successful experiments and encourage researchers to structure and share this kind of data. This can be the pillar to improve and upgrade the synthesis strategies and reduce the e-waste (data that is discarded without any gain by the effort, time and resources spent). This approach is not focused on establishing proper process–structure–property relationships but rather enriching the knowledge base and providing a chemical intuition on how to approach strategically the parameter design and avoid repeating unsuccessful experiments. Optimization algorithms that guide space exploration can be the engine for the predictive models, i.e., a NN to extrapolate the established materials correlation to find new knowledge.

Moosavi et al., performed genetic optimization by using thirty new experiments for each “generation”, while optimization was initiated with the more diverse individuals to ensure the non-biased exploration of the chemical space. Random forest and bootstrapping were used with a total population of 200 trees to model the synthesis parameters, using ‘out-of-bag’ technique for validation. The size of the reactor, the purity of reagents, and other relevant parameters influence the experiment. It has been indicated that access to more data would enable the machine learning algorithms to improve the filtering of inhomogeneities and in choosing the most influential variables, thus effectively reducing the number of experiments [103].

The ultimate target is to achieve and integrate experiment, theory, and computer simulation all in one, to automate design and create innovative materials iteratively by selecting the parameters for the next most promising experiment. Such an integrated system has been used in manufacturing processes for CNTs to speed up their development for application in avionics. For a different case study, in the University of Berkeley, they used SoA theoretical tools and a database of 66,000 instances of crystalline compounds and 500,000 nanoporous materials [104].

In another case, GAs have been utilized to optimize the physical and functional properties of metal-organic frameworks (MOFs), more specifically referring to functional chemical groups and pores. Collins et al., modelled the CO_2_ uptake capacity in order to maximize MOFs capability to reduce the environmental footprint. Thus, 141 experiments under conditions relevant for post-combustion CO_2_ capture were used. In this case, screening of the possible MOFs configurations varied among 1.65 trillion structures and high throughput screening was performed by the GA. A total of 1035 derivatives MOFs structures were identified to offer exceptional CO_2_ uptake >3.0 mmol/g (at 0.15 atm and 298 K), while several known structures were optimized; MIL-47 entitled MOF structure demonstrated an increase in CO_2_ adsorption by more than 400% (4 mmol/g). Collins et al., in their study developed a customized GA to select the functionalization of the parent MOF as the mechanism to derive optimized functional and/or physical property by searching a small chemical space (CO_2_ uptakes in 141 experiments), in order to satisfy time-efficient computing. The outcome of the study was to optimize more than 20 known MOFs. However, with complex reaction paths, reaction yield can be the bottleneck for scientists, even though the population of the optimized generations provides multiple solutions for the optimization problem and selecting the synthetic targets. One interesting capability of the model was the capacity to be exploited to similar chemical real-life problems, such as the development of improved covalent organic frameworks or polymer porous networks [105]. This is an additional showcase of materials design and high-throughput screening capabilities of generic algorithms.

Other than GAs, the Bayesian-assisted process optimization has been employed in a representative optimization case of an annular microreactor synthesis, which was employed to upscale the synthesis of antibacterial nanoZnO to a production scale of 1 kg/day. In this case, mining of the available data was focused on commercial antimicrobial and experimentally synthesized nano-ZnO to increase the productivity scale in less than 100 experiments [61]. Moreover, the establishment of process–structure–property relationships revealed that nanostar and nanorod architectures are related to the assembly of nanoparticle precursors, and the antibacterial properties were owed to anisotropy, surface area, and particle size distribution.

Other approaches for materials design have been envisaged with dynamic parameter design based on back propagation ANNs, by combining a single step optimization procedure without the need for the assumption/estimation of an adjustment parameter. Modelling is performed including features and noise, design and signal descriptors. Hyperparameter tuning was implemented to optimize the modelling accuracy (hidden layer number, momentum, number of neurons, learning rate). Latin hypercube sampling was employed to estimate the parameters related to the contribution of the signal and noise in the dataset, and afterwards sequential quadratic programming was employed to efficiently model non-linearity of data. It was shown that a two-step procedure, like Taguchi parameter design (maximization of characteristic to signal ratio and adjustment of the slope), requires an adjustment parameter in order not to settle in a local optimum solution within the limits of the dataset, while it is assumed that the signal data follow a uniform statistical distribution, which almost always cannot represent a real-life problem. Thus, in the relevant example of plasma enhanced chemical vapor deposition method (PECVD), the optimization of the process was shown to be challenging using a Taguchi optimization method. The proposed method by Jung et al., can optimize the dynamic design parameter regardless the existence of the adjustment parameter and other relevant shortcomings of the Taguchi method. The optimization was achieved only by the inclusion of characteristic signal, noise, and expected loss in the model to solve the non-linear problem [106].

Electric discharge machining (EDM) is another industrial process. This process is used to reshape materials, where several methods have been used in the past for optimization, namely, factorial analysis, analysis of variance (ANOVA), response surface method (RSM), regression, and Taguchi analysis. Taguchi analysis can be used to find the best specified parameter, within the levels of the process data, while ANOVA and regression analysis can be used to describe for instance the electron wear (EW) in die-sinking in the steel material of EDM, and maximization of the material removal rate (MRR). Emerging optimization techniques based on ANN, GA, PSO, and simulated annealing (SA) have been promising to make actual proceedings in the field, as well as to find and predict a verified improvement in the machining process with high accuracy. Even evolutionary algorithms demonstrate some shortcomings. SA requires only one population to converge and a higher computational cost considering that it can easily fall among a local optimum, and takes more time to escape, while also GA may face difficulties in finding a universal solution. Thus, Majumder et al., in their approach used a hybrid algorithm consisting of three evaluators in parallel (GA, SA, PSO), each of which use a back-propagation NN, while the outcomes of the optimization process were validated by a trial-and-error approach. In this case, PSO could efficiently predict the optimum number of neurons in the hidden layer/-s, while both SA and GA require more computational power to converge [107].

To conclude with, promising progress is realized in this field. A main occurrence that is faced by the scientists is the high demand for computational resources needed to guide through the design parameter space. In this aspect, moving towards a more goal-oriented approach could be the key to identifying the best candidate (nano-)materials since convergence time is highly proportional to the number of search parameters. In the latter case, some limitations can be confronted by missing data regarding materials properties or synthesis parameters, which can be bridged by utilizing simulation data and obtaining dependable alternatives when it is not possible to repeat experiments or there is limited access to characterization facilities.

### 3.2. Utilization of Synergistic Modelling-Simulation and Combination of Ensemble Machine Learning Algorithms for Selection of Materials and Process Parameters

Machine learning operations are purely data-driven, and this feature has been proved in recent years to provide practical solutions in material science field by providing a set of decision rules, which correspond to materials physical and chemical properties. The establishment of these structure–property relations have the potential to feed the simulation models in the future [12], and reduce the required cost due to the lower level of complexity, but currently accuracy has been the main concern, which cannot still reach DFT capability.

Most of the manufacturing and engineering sectors require optimization of process parameters and adaptation. For instance, to support the expensive and demanding polymerization processes. Simulation approaches include the use of FEM and computational fluid dynamics (CFD) for determining the optimum set of design parameters with high precision, which often vary from ten to several hundreds. However, FEM and CFD in most cases have been demanding in regard to the simulation time to solve those non-linear problems and there is plenty of room for achieving shortcuts by using, e.g., surrogate-based optimisation (SBO). Surrogate machine learning models are characterized as approximation meta-models, such as polynomials, ANN, kernel-based, decision trees, and can provide approximations of high-fidelity models [108]. To better explain the term, if the machine learning model is supposed to replace a time-consuming physics-based simulation, it is called a meta- or surrogate-model [53]. The input by these models is iteratively optimized by simulation evidence as a measure to reduce the bias introduced by the approximation parameter. Such is the case for the ANN modelling in a 50-dimensional space of the shear angle of 24,000 composite fabric elements. Such a model required only 20 updates, while the efficiency of the surrogate model can be further optimized by using, e.g., Latin hypercubes for the principled selection of initial samples or exploration/exploitation tradeoff in regions with higher uncertainty (less experimental evidence) [108].

A similar hybrid machine learning and simulation SBO approach has been realized. Zhou et al., used reinforcement learning to optimize the chemical reactions. A deep neural network (DNN) architecture was used to iteratively record the predicted result and re-design the next experiment. The “Deep Reaction Optimizer” model succeeded its processors of SoA blackbox methods, such as covariance matrix adaption–evolution strategy (CMA-ES), in regard to the complex development of chemicals based on simulations and experiments, which were reduced by 71 required steps compared to the current SoA method. Several probability distributions were strategically used to further explore the chemical space, which was finally combined with microdroplet reactions. In this case, optimization was achieved in 30 min and included the outcome of four reactions with different mechanisms. The deep reaction optimizer was deployed for the experimental setup of silver nanoparticle synthesis. The optimization target is the nanomaterials activity of the absorbance at 500 nm, which corresponds to Ag nanoparticles with diameter of 100 nm. In their study it was shown that the model is extendable to bulk-phase reactions, while another functionality contained the ability to learn while optimizing the development process. For the training of the deep reaction optimizer, two model reactions were used; the Pomeranz–Fritsch synthesis of isoquinoline and the reaction of the Friedländer synthesis of substituted quinoline, while further reaction parameters (voltage, pressure, flow rate) were mapped using a Gaussian process for optimizing the reaction yield [109]. In this paradigm, the main uncertainty is related to the use of the chemical reaction thermodynamics which are involved in the reaction mechanisms, which can have a detrimental effect to the what if analysis employed in a similar optimization problem. For the synthesis of nanomaterials, both the chemistry (state of bonding, interatomic potentials, etc.) of the precursor and the dynamics of the reaction (temperature, pressure, flow rate, etc.) can provide descriptors that enable the generalizability of the derived models.

On another occasion, Pt nanoparticles were studied by Lansford et al., in order to evaluate their efficiency in CO capture as indicated by their properties. DFT simulations were used to accurate generate the absorption sites in the IR spectra corresponding to the C-O and Pt-C bonds, with one intended purpose to reduce uncertainty and noise in the signal of experimentally acquired IR spectra. DFT was used to predict the frequencies at low adsorption of CO, which generated a dataset of 1090 unique intensities and frequencies, including also structural information about CO coordination environment and binding type (bridge, atop, threefold, and fourfold sites). Those data are used to produce a second dataset based on surrogate models, which are physics-driven using forces and dipole moment in modelling under harmonic approximation. Subsequently, adsorption sites were quantified using as key identifiers the binding-type and the generalized coordination number (GCN). The experimentally produced IR data were fitted using probability distribution analysis. Then, the authors derived a closed-form solution, which is correlating the Wasserstein distance parameter with the softmax activation function, which is used to train NN machine learning models that belong to “ensemble” family. These networks (two separately, termed as structure surrogate models) were trained to recognize the probability distribution functions of GCN and binding type from the simulated spectra, and the error was well quantified by the NN models for the case of CO. The model demonstrated ability to generalize to NO probe molecule, as well as in predicting the adsorption using both clusters and Pt nanoparticles. It was shown that the spectroscopic signatures of Pt–CO and C–O were correlated to the coordination environment, while the case study was benchmarked using simulation and experimental validation [93]. This is an interesting approach to describe the adsorption capabilities of Pt nanoparticles; however, the approach is highly dependent to the adsorption sites, which is nanomaterial and particle specific. In the simulation, particles are generated in a structured way, but in reality, the statistical distribution may vary, whereas the cluster formation and assembling is application dependent, i.e., on how this technology is deployed in a device.

The motivation for using mathematical and physical theories in simulation is to extrapolate the properties of (nano)materials and select among the best candidates. Ghiringhelli acknowledged some basic considerations for the descriptors in his research; low-dimensionality, unique connection with the material, and very different descriptor values are suitable for selecting and characterizing different materials. Testing a large number of molecules with high-throughput simulation, to sort the best candidate configurations by their correspondence to the need of a specified application, such as selective adsorption, removal, stability, catalytic properties, affect the computational cost, which is a significant drawback. Even in the case of quantum chemistry, DFT is limited by the few candidate structures that can be modelled by this method. Especially, in the case of MOFs, data-driven methods seem at this time to be more efficient to tackle the optimization task of this real-life problem [63]. Again, in the case of data-driven models, machine learning computation time is affected by the number of descriptors, and still high level multi-dimensional models are demanding in time and cost resources to achieve the evaluation of the targeted property.

Similarly, Zhou et al., proposed an effective way to screen the predicted crystal structures of zeolites, MOFs, COFs, ZIFs, PPNs, and experimentally synthesized zeolites and MOFs. The materials selection was guided by the Xe/Kr separation at room temperature in a sample consisting of 670,000 structures. The authors used a hybrid method to avoid expensive computations, by combining simulation with machine learning. They used several descriptors for the aforementioned (nano-)materials, such as the largest dimension of spheres, surface area, crystal density, void fraction, as well as a new descriptor “Voronoi energy”, named after the respective model. The authors found that Voronoi energy was the descriptor with the higher influence in predictions, and could be regarded as the fingerprint of these materials. A high dimensional model was developed using a sample of 15,000 observations and random forest regression algorithm (in the family of decision trees) with the aim to predict xenon selectivity over krypton. The model was then deployed on 655,000 materials to predict their selectivity. Molecular simulations were used to refine the predicted outcome of the random forest model, granted that the prediction was promising for the effective screening of the Nanoporous Materials Genome, so that molecular simulations were used for 20,000 out of the 670,000 available structures. A significant outcome was shown to be the fact that many candidate materials from the database could outperform the Xe/Kr separation properties of current state of the art materials; JAVTAC zeolite analogue and KAXQIL calcium coordination network were identified as the most promising materials for this application. For Xe/Kr separation, no geometric parameter could be directly correlated to selectivity, which would possibly simplify the recipe for highly separating materials. Future expansion of the hybrid model could be implemented in a similar manner, and highly efficient high-throughput screening can be performed by modelling and predicting other material properties, such as gas sensing and storage, catalysis, and drug delivery capabilities [110].

In the same research field, the Nanoporous Materials Genome Center is highly active, and MOFs and zeolites are studied about their electronic structure utilizing Monte Carlo sampling methods and the theoretical analysis is taken advantage to establish a screening workflow. Properties that are used for material selection contain carbon capture, catalysis, phase separation, and gas storage [17].

In the study of Zhou et al., emphasis was focused on the characteristics of active centers to determine the adsorbate binding strength, which till now are uncertain. Even the adsorption free energies of –O and –OH functional groups/free radicals can be directly correlated with OER activity, the computational cost does not enable a systematic study. Non-local density functional theory (NLDFT) was used for the measurement of pore size and distribution in the COFs, while the electronic structure and the optimum metal-coordination environment was shown to be derived in case of Ni-COF electrocatalyst, which was validated by X-ray techniques. The free energy relationship of metal COFs was shown to be dependent on the metal used using DFT, more specifically Zn < Cu < Fe < Ni ≈ Co, which was experimentally validated, except for the case of Co–COF and Ni–COF catalytic OER activities. In the outcomes of this study also two promising candidates (experimentally unexplored) were proposed to be synthesized in the future studies; Fe–N_3_O and Co–N_2_O_2_. In order to access more sustainable calculations, effort was spent on the identification of alternative descriptors with machine learning to simplify the modelling of OER activity of catalysts. The study utilized 100 structures with 23 unique features. Electronegativity, ionic radii, electron affiliation energy, and modelling was performed to evaluate the first ionization energy of metals. In this case, the regression algorithm used was the gradient boost (GBR) to give more insights and describe the gradient of the free energies of oxygen and hydroxyl radicals. A high correlation was shown to govern the relationship of electrostatic interaction and adsorption energy of intermediates. Final screening of material features showed that four intrinsic factors are only required to describe the OER activity of the model catalysts metal-N_x_O_y_, while the predictions could be well-correlated with DFT predictions [111].

Especially, in the field of MOFs and COFs, a wide variety of approaches and descriptors has been utilized to confront engineering and application oriented problems. The main issue is that it is still a grey field on how research should be oriented, and a better organization of the computational research should clearly provide guidelines, i.e., which are the parameters with the higher sensitivity compared to the study objective, and which parameters should be characterized or modelled by the researchers to reach dependable conclusions. Currently, there is a lot of sparsity in the machine learning approaches which could be connected to the researchers’ backgrounds as well as accessibility to characterization and computational facilities. This limitation can be efficiently overcome by structuring materials data and sharing them in open repositories, as well as by establishing roadmaps and community standards.

In another interesting field of carbon nanomaterials research, Xiang et al., performed machine learning simulations in a molecular dynamics HTMD environment at high-throughput in order to tailor and optimize CNTs structure in regards to mechanical performance. The structural features were studied, namely number of walls, crosslink density, chirality, diameter, and it was revealed that armchair configuration, as well as the minimization of diameter lead to optimization of nominal tensile strength. Moreover, it was found that a small wall number with a higher outer diameter of 43.39 Å is beneficial when it is combined with a high crosslink density of the adjacent walls. More specifically, for the armchair-type CNTs that consisted of five walls, the nominal tensile strength varied between 58 and 64 GPa, nominal Young’s modulus between 677 and 698 GPa, effective tensile strength between 65 and 71 GPa, and effective Young’s modulus between 730 and 754 GPa. SOMs were used for the visualization of the results, and it was shown that the crosslink density negatively influences the mechanical properties when the number of walls is high, due to the lower density in the outer walls [112]. In this case, the authors offered a robust case establishing a useful link between the chemical structure and performance. What is missing is the link to process data descriptors in order to verify the results. This is often the case owing to the multidisciplinary needs of the field.

Arabha and Rajabpour studied the elastic modulus and thermal properties of carbon-based 2D nanostructures, which is conventionally implemented with MD and DFT approaches. The drawback of MD is that the output is strongly correlated to the accuracy of the estimation of interatomic potentials, while DFT can offer this information only at a high cost. In their approach, the elastic modulus and thermal conductivity of nitrogenated holey monolayer graphene was achieved using machine learning passively fitted interatomic potentials (MLIPs), which depend on the output of non-equilibrium MD simulations. In the one case, the thermal conductivity of graphene lattice was measured at 85.5 ± 3 W/m-K, which was accompanied by an effective phonon mean free path of 36.7 ± 1 nm. Then, the uniaxial tension was simulated for the measurement of elastic modulus (390 ± 3 GPa), ultimate strength (42 ± 2 GPa), and fractural strain (0.29 ± 0.01). Thus, the machine learning approach was shown to be effective and accurate utilizing classical MD simulations. The model was able also to regenerate the phononic properties with remarkable accuracy close to the DFT approach, while it was revealed that thermal conductivity is length-dependent. Moreover, by training MLIP, the point defect configurations were detected, which gave insights regarding defective structures and their mechanical properties. For the case of 2% defects, elastic modulus declined by 5%, while fractural and ultimate strength were also reduced (by 40% and 15%). However, even this MLIP approach was shown to require a high computational cost [22].

Even in the case of surrogate models and the combination of machine learning with simulation models, the computational cost remains a barrier to realize the benefits of computational modelling. Quantum chemistry can be in the center of the solution to this problem, offering accurate and faster quantum-mechanical measurements of electronic effects and contributions which can be used as an alternative to establish structure–property relationships. The use also of machine learning potential has been proposed as a promising solution.

The combination of machine learning and quantum chemical calculations at low level of theory has been realized for the precise simulation of hundreds of thousands of molecules, which enables the avoidance of expensive electronic calculations at the highest level [46]. The foundation of atomic-scale modelling has been mainly focused on the measurement of cohesion energy of electrons or atoms, as it is connected to the majority of the occurring phenomena, and with materials properties as well. Few assumptions can provide an accurate picture of the effects that govern the materials behavior, which is predicted accordingly. A common case in engineering field is the modelling of the diffusion rate of elements, using the simulation of material applications exposed to extreme conditions, where actual testing is not possible. Approaches such as DFT require high computational cost even for supercomputers; depending on the number of electrons in the material requires powerful resources to compute the material energy in a reasonable timeline. Another example is the prediction of the melting point of a material, which may take up to thousands of hours to converge. Thus, quantum computing holds a lot of promise for future excellence in the field. Currently, efficiency is the key word to enable faster computation. In the DFT approach, the dynamical mean field theory can address the efficiency issue, especially for Coulomb interactions on a single particle structure. Similarly, density matrix renormalization and quantum Monte Carlo methods have been emerging to model i.e., high temperature superconductivity. Phenomenological models should still be adopted to describe the behaviour of specific materials to maximize their impact [3].

Another proposal is to use deep tensor neural network models (DTNNs). It was shown that DTNNs can model accurately chemical environments with atom-centered bases (rotation, translation, permutation) and molecular energies using for training the reference DFT calculations obtained by GDB database, using one dataset consisting of 7211 molecules, and another with 133,885 molecules, containing compositional data the contained elements (including heavy atoms), MD trajectories, data of atomization energies, and exchange correlation potential. Due to its functionality to construct the model recursively using pairwise distances, the model can also be employed to generate “alchemical” reaction paths. DTNN in this study by Schutt et al., demonstrated an advantage compared to previously developed ANNs; there was no need to develop separate ANN models for each non-equivalent many-body system, due to the common (and not fixed) quantum chemical bases that effectively describe the high-order interactions and which atom interacts, which does not require to select manually symmetry functions and adapts to each scenario due to the data-driven character of DTNNs. These advantages can be promising to gain quantum-chemical insights with the referred model to achieve an accuracy of 1 kcal/mol even for intermediate size molecules throughout configuration and compositional space, for instance local molecular chemical potentials, relative isomer energies and aromatic ring stability, and the detection of molecules with peculiar electronic structure [113].

In another case, machine-learning potentials (MLPs) were used to reduce the computational cost of modelling the output data from first principles quantum-mechanics methods by Chen et al. The MLP based Monte Carlo and MD simulations were assisted by a sampling methods/packages (TINKER molecular modelling software; atomic energy network ANN potential package, ænet; and large-scale atomic/molecular massively parallel simulator, LAMMPS) to increase efficiency without costs in accuracy when describing complex and large systems, while scalability was more effective using the ænet–TINKER interface (limited to shared memory systems). Two specific instances were described using the combination of machine learning and simulation for modelling: the equilibration of nanostructured battery materials and the study of diffusion in liquid water. Both interfaces were shown to provide a sustainable solution to large-scale MD simulation taking advantage over parallel computing capabilities by enabling a single step computational time at (sub-)millisecond scale; the shared-memory ænet—TINKER showed high level of parallel efficiency using a single computer node, while ænet—LAMMPS interface demonstrated 80% efficiency at scaling on multi-mode architectures (up to 100 computer cores), both demonstrating an accuracy close to first-principles methods [114].

## 4. Selection of Nanomaterials Tailored for Improvements in Quality of Life; Human and Environmental Health

The challenging task of effective and safe nanomaterials design is the most common bottleneck to wide commercialization. A shortcoming nowadays in commercial nanomaterials and nano-reinforced composites is about the documentation of their safety datasheet, which is not always available or comprehensive. Major steps and efforts have yet to be realized for establishing a thorough characterization of nanomaterials, commonly accepted nanodescriptors, and high-throughput in vitro methods to assess toxicity and relevant endpoints. Another major discussion in the field includes the current in vitro methods performed to animals which have provided inconsistent results when deploying the studies to human cells, due to differences in human pathophysiology and biology, which can be resolved by using in silico methods [45].

In case of bioresearch fields, there is also an ethical aspect to consider. Performing experiments on animals is lengthy and costly, while in 2006 Registration, Evaluation, Authorisation and Restriction of Chemicals (REACH) suggests the adoption of alternative research methods, such as in silico [40,115]. In silico methods for nanomaterial design have been emerging with quantitative structure–activity relationship (QSAR) models, providing implications in drug and molecular design of small sequences of organic molecules. These models, however, are limited by the amount of data when examining the case of new nanomaterials, while the identification of suitable nanodescriptors (empirical, experimental, structural) is another main challenge in predicting the behavior of complex structures. To address this challenge, the descriptors that will be identified should address the interoperability need of these models, namely to represent adequately the physicochemical behavior (i.e., zeta potential, relaxivities, bandgap) and describe the structural diversity (i.e., size, shape). These descriptors are essential to train the intelligent machine systems to identify how the features critical to nanomaterials safety and stability are affected and perform virtual screening. For instance, protein corona fingerprinting, which describes how serum properties are on the surface in case of cellular models. In this direction, two descriptors with the ability to generalize the predictions on nanomaterials have been identified by Yan et al., namely the Pauling electronegativity and Delaunay tessellation approach, which simulates the surface chemistry of a nanomaterial [48]. Consequently, feature selection is an essential methodology to improve the ability of predictive models in the tricky field of both environmental and safety monitoring by predicting precisely the biological effects and mechanism upon exposure to nanomaterials. Establishing proper structure–property–activity relationships depends on descriptor selection, which has been used in numbers, and often vary amongst the scientific community. Thus, this methodology has found application in numerous studies by using the sensitivity analysis to identify non-correlated descriptors. Feature selection, first of all, can be used to tackle the overfitting problem in machine learning models and provide higher generalizability potential, considering that universal descriptors of (eco-)toxicity are yet to be realized. A second benefit corresponds to the improved possibilities to perform mechanistic basis for the model by the experts in the field [57].

Another emerging need is about binding the outcomes of both in vivo and in vitro studies to support regulators for policymaking, which could be achieved via the development of in vivo machine learning models. This is an example of the approach towards this direction by US National Institutes of Health and Environmental Protection Agency (EPA), which tries to collect and provide reliable information based on an accepted suite of cell-based assays, which are used for mapping the toxicity profile of chemical compounds. Similarly, it can be possible to establish the broad toxicity profile of nanomaterials using also machine learning modelling. Already, the development of such high-throughput methods is realized by focusing on short-term milestones for measuring the interactions of commercial nanoparticles with plasma proteins, while there are coordinated efforts for the establishment of enhanced monitoring methods to enable tracking of nanoparticles during their residence in the human body. Other orientations contain the development of information sharing mechanisms, which has been a multidisciplinary challenge to be address by computer scientists, experimentalists, industry, and regulators. Already, an increment in in vivo data has been evidenced to describe nanoparticle effects, while also relevant work is being performed in database development to support the predictive modelling for nanoparticle corona formation in different environments, the elucidation of mechanisms to describe the effects of nanoparticles which enter into cells as well as toxicity mechanisms, to name a few, apoptosis, free-radical production, and genotoxicity [45].

By establishing data-driven connections between nanomaterials and, e.g., the immune system, it possible to drive material design and ensure greater safety. The major battlement remains related to the access to clinical applications data, and actual paradigms such as immunotherapy [40]. For example, it is known that it is easier for the smaller nanoparticles to penetrate the cellular membranes and interact with organelles [27], but other individual features also affect the nanomaterial behavior and interactions. Digital infrastructures and databases are expected to bridge this gap of knowledge and support the discovery of nanomaterials that can embrace the activation and inhibition of therapeutical mechanisms based on their individual features; though it required a massive effort to create general rules [37]. Especially, systems and bodies connected to machine intelligence and IoT (Internet of Things) or IoB (Internet of Bodies) depend on the connectivity of sensors with detection capabilities to enable the high-throughput transfer of data via for instance radio-frequency identification devices for learning and adapting decisions, also termed as informed guesses, that are related to the components qualitative and quantitative features [9].

Many research groups approach the machine learning modelling of real-life problems with consciousness about the importance of the amount of data to represent the problem with quantitative and deep structure property relationships and perform high throughput screening to identify the most promising candidates [21]. A representative example is related to the work performed by Courtney R. Thomas’s team who developed a high-throughput recommendation system for engineered nanomaterials to predict the nano- and eco- toxicology, safety, environmental impact, and relevant properties [116].

Thus, nanoinformatics utilization in the Big Data era for the prediction of (eco-)toxicity of nanomaterials is emerging as a science to increase confidence and support the commercialization of innovative nanotechnology [52]. Nanoinformatics is a crucial field for providing methodologies to categorize nanomaterials based on their physicochemical behavior and (eco-)toxicological properties to increase the knowledge base and support the development of read-across models (Figure 4). Several strategies, such as that provided by ECHA, have an enabling character to allow the in-silico exploration of nanomaterials toxicological endpoints based on an experimental data-driven approach, which can support in the future regulatory testing [117]. What it is more important with data and nanoinformatics is that the machine learning tools and QSAR/QSPR models for the correlation of nanomaterial properties with toxicity, exposure, and hazard assessment can support the policy adaptivity to new nanotechnologies in the future upon standardization [57]. Thus, design procedures can be used to maximize nanomaterial utility while there is compliance with European Commission and global strategies to replace, reduce, and refine (3R principles), which aims to reduce adverse biological effects and support ethical science by reducing the tests performed using animals [47,49].

### 4.1. Machine Learning Modelling of Biological Effects of Nanomaterials

What has been evidenced regarding the conventional QSAR modelling is that even the access to data of different nanomaterials may be sufficient, though the diversity in descriptors in terms of both uniqueness of the identified exposure problem and the availability of common descriptors for different nanomaterials. Typical descriptors are about the physicochemical properties, composition, surface chemistry, and structure, which can be used to optimize protocols for physical/chemical description by establishing the principles for the interaction between different material classes and living organisms. This shortcoming is limiting the ability to identify universal descriptors and perform proper virtual screening of nanomaterials, as well as their suitability to characterize various properties, such as physicochemical, biological, or structural diversity, while dynamic changes and interaction with the immune system can alter the aforementioned nanomaterial features, and thus their behavior, i.e., by biotransformation in the body as in case of protein crown adsorption [37]. Machine learning can return many benefits, such as fast computational speed and high accuracy, while it is resistant to interference (anti-isomorphism), without the need to utilize complex mathematic formulas, and with the ability to provide a generalized solution to a specific problem. Feature engineering can be used to filter out the critical parameters that are important, in order to give more insights about nanomaterial related immune responses.

A common problem in machine learning is the modelling capabilities of individual algorithms, hyperparameter tuning, and access to high quantity and quality of data, which may limit the model complexity. This means that the predictions and functionality of the models are limited by the domain of the properties that are fed into the model, and predictions beyond that domain would lead to poor results. This highlights the need for using datasets that cover the chemical diversity of the studied nanomaterials, while molecular and biological descriptors are sufficient to describe the expected behavior for the interaction with i.e., human cells. Moreover, experimental validation of the models predictive ability can enable the utilization of such predictive models by regulators and scientists, since machine learning can establish useful tools to study the molecular mechanisms and describe nanomaterials biological activity like in case of QSAR models [118].

Another feature that may affect the predictive ability for a toxicity endpoint in many cases is toxicological model bias, since the introduction of weights is often missing. More specifically, since missing a toxic instance will have a negative and non-reversible impact, which should be prevented by design; for instance, by introducing weights to perform class rebalancing or introduce a cost-penalty factor in wrong predictions. Furthermore, to tackle the issue of the bias induced for the prediction of the majority class of data, resampling techniques can be applicable such as the SMOTE (Synthetic Minority Over-Sampling) technique [36]. Subramanian and Palaniappan used the SMOTE method to address this class skew problem and reduce the training bias by creating synthetic toxic instances of the minority data based on the data features of the minority class, while the other classes are not affected at all. Utilizing in vitro features in a principled agnostic approach led to a total accuracy of 96%. Hyperparameter tuning and trimming of the feature space were employed, based on intrinsic and extrinsic physicochemical properties, together with periodic table features, which are exclusive of in vitro parameters, e.g., cell type and line and assay method, for multicollinearity [119].

In the field of carbon-based nanomaterials, Gernand and Casman in their meta-analysis studied the pulmonary toxicity effects of 17 CNTs types on guinea pigs using random forest algorithms. They used the structural features of CNTs, the concentration of metallic impurities, the dose and duration of exposure, and other features obtained by the observations in laboratory. The authors could effectively predict the toxicity endpoints, i.e., number of polymorphonuclear neutrophils, lactate dehydrogenase, number of macrophages, and total protein concentrations, with a confidence of 0.88–0.96 as indicated by the r-square value, where the parameters of the presence of metallic impurities, surface area and charge, agglomeration, as well as nanotube length and diameter were shown to be important modulators of CNTs biological effects [120].

In another study, CNTs were modelled about their potential genotoxicity, which is an important aspect of their behavior in order to enable their exploitation in numerous applications. A fully mathematical model was developed by Kotzabasaki et al., to describe their toxicological endpoint in terms of genotoxicity considering the fact that it is a well-studied field in literature and accessible data demonstrate completeness in accordance to REACH requirement. Both in vivo and in vitro studies were involved to develop the model, while two cheminformatics workflows were executed using various SoA machine learning methods, which were validated using multiple performance criteria. Some promising models were developed using the random forest algorithm, as well as linear regression. Both models were able to model properly the toxicity effects using only a few descriptors, and more specifically Zeta average, length, and percentage of pure carbon. In case of linear regression improved results were obtained when polydispersity index was included. These descriptors have been highlighted also by literature about their connection to genotoxic hazard potential of CNTs, and especially larger length was reported to induce in vivo toxicity and resulting to cell death by a reactive oxygen species generation mechanism. Further insights are considered to be obtained if the descriptors include information about extrinsic properties of the surrounding medium, such asionic strength, serum proteins, and pH [115].

Gonzalez-Durruthy et al., studied a more specific case about the biological impact of CNTs on the mass flux of oxygen in mitochondria. Non-significant respiratory effects were observed when CNTs do not exceed the 5 µg/mL concentration, which corresponds to a typical pharmacodynamics criterion of NOAEL. For this purpose, the data obtained by Raman spectroscopy and subsequent SG transformation were used to define new nanodescriptors and realize the dosimetry correlation to adverse effects on mitochondria. Raman analysis is expected to provide real-time feedback based on the applied experimental conditions about the oxygen mass flux, which can be influenced by the presence of CNTs. This approach can provide a sustainable methodology to analyze CNTs effects by generating a massive amount of raw data to support decision-making at regulatory level in the field of biomaterial science. The dataset used for this study is available at FigShare repository, including the following parameters: 32,940 cases of mitochondrial oxygen mass flux *J_m_(O*_2_*)* in the presence of CNTs, and 34 features, including various chemical modification of CNT, Replicate, Solvent, and time for various types of CNTs, such as single walled, mixed single walled/double-walled, and multi-walled. PT machine learning regression models were established using R language and BioCAI HPC platform (available in github repository), and more specifically NN, linear regression, as well as random forest regression algorithms were employed. The best performing model in each case was selected based on the prediction efficiency in the test dataset based on the maximization of regression coefficient and the minimization of RMSE (root mean square error) values after proper tuning of model hyperparameters in order to fit the RRegrs methodology, while the overall best performing model has been deployed online to serve the identification of the mass flux of oxygen in presence of CNTs for the mitochondria case [121].

Within the family of CNTs so far, we navigated through three different risk assessment studies, where three different approaches were utilized. A significant difference that can affect and turnover the result of each study is the purity of CNTs. For instance, the simulation of CNT structures leads to pure nanotubes, unless otherwise identified, while experimentally they may contain traces or higher concentrations of the catalyst, chemisorbed ions if they are functionalized, and so on. This case is not limited to CNTs but can be extended to every nanostructure where a data conflict can be confronted. The main point is that if the experimental data that are accessible refer to the effects of pure nanomaterials, the documentation of the toxicity effects can lead to unreliable structure–property–activity relations. Thus, the researchers should make sure to be precise in documenting data (not limited to purity example) in order to be reusable in future studies and reduce conflicting predictions of different machine learning algorithms.

In the case of graphene, studies often demonstrate conflicting conclusions, while it is thought that size is the main driver for cytotoxicity, followed by time and dose dependent exposure effects which have been evidenced. Furthermore, surface modification, oxidation state, and surface charge were shown to be involved in the graphene stability changes and act as a main driver of toxicity in bacteria and human colorectal adenocarcinoma cell line HT-29 compared to graphene in its pure form. Moreover, PEGylation or treatment of graphene with Pluronic acid has been shown to enhance the biocompatibility of graphene, while in case of graphene oxide cytotoxicity effects are lowered if the surface charge is negative. In addition, factors such as cell morphology and type, as well as organ type led to different toxicity response profiles when exposed to graphene related types; for instance, breast cancer cells are not affected at all when exposed to PEGylated graphene oxide, while lung cancer cell have a 30% viability. In the case of liver cancer, it was shown that cells are damaged at the DNA level. In this direction, Bayesian networks and random forest models were considered promising to manage data mined from literature and databases in order to develop a predictive modelling framework about the factors affecting cellular toxicity of graphene and deliver a holistic hazard ranking assessment of graphene, as well as support rational nanomaterials design, environmental, and health policies [122].

Apart from carbon-based nanomaterials, numerous studies have focused on the effects of metal-based nanomaterials. More specifically, de Pablo et al., studied GNPs by establishing QNAR models using random forest machine learning algorithms in order to perform rapid in silico screening of newly designed structures. The virtual representations (vGNPs) were generated using an in-house “GNPrep” program, which is also placing the surface ligands in a randomized manner about the density representation of Au-S bonds in the surface and the data were stored as Protein Data Bank (PDB) files. Hence, 100 representations were used for the descriptor calculations for each GNP to avoid possible instabilities. The vGNPs represent diversity in the surface chemistry as derived by nanocombinatorial chemistry technique, which is used to attach on the surface different small organic molecules. Optimization of virtual structures was performed by energy minimization method (such as the amber forcefield method) and the main descriptors that were identified correspond to the Pauling electronegativity and interface geometry. Moreover, other known descriptors of hydrophobicity and zeta potential were used to adequately model the bioactivity of the nanoparticles, such as cellular uptakes, GNP- enzyme bindings, and ROS inductions [3].

Yan et al., used 147 nanoparticles synthesized by nanocombinatorial library approach to generate images of their nanostructure, and used a two-step annotation process. First, the simulation is used to generate the virtual representation of the nanoparticle. Then, the nanostructure image is generated based on the virtual representation by the first step. Four basic nanoparticle attributes were utilized in the creation of the virtual image, including the size of nanoparticle core, the type of core material, the surface ligand density, and the chemical structure of surface ligand. A CNN was developed based on LeNet architecture, which implements Keras and TensorFlow. Zeta potential, superhydrophobicity, protein adsorption, and cellular uptake were used as descriptors. The model architecture consisted of the feature extractor, which corresponds to the first four pairs of convolutional and max-pooling layers, and the predictor, which is consisting of two fully connected layers, resulting in an agreement of the training and validation loss. In this case overfitting was avoided using data augmentation and dropout regularization. The dataset was consisted of 12 platinum, 12 palladium, and 123 GNPs, with a size less than 10 nm in most cases, which is similar to the size of proteins. A class activation map was used to observe the regions in the image, while the deep CNN was shown to act as an information distillation pipeline, by using raw data to filter out irrelevant information and effectively magnify and refine information useful for predicting the in vivo fate for nanoparticles while entering cell lines [29].

Trinh et al., developed a nanoQSAR model to provide a time and cost-efficient predictive method for the assessment of toxicity of gold and silver nanoparticles using data mined from 63 published articles. Data were curated and resulted in a dataset with 2005 observations and 31 descriptors, while missing data were completed by using manufacturer references. PChem scores were used to evaluate data quality and completeness to subsequently develop random forest and support vector machine classification models, which showed that surface charge, core size, dose, and cell lines were influencing the toxicity of the studied nanoparticles [123].

Joyita Roya and Kunal Roy worked on the establishment of nano-QSAR models for metal oxide nanoparticles towards RAW 264.7 cells and the release of lactate dehydrogenase from the cell as an endpoint, which can be interpretated also in non-investigated nanoparticles. 25 different nanoparticles were studies, amongst them, Ni(OH)_2_, ZnO, SiO_2_, AlOOH, CeO_2_, and TiO_2_. In their study, they developed partial least squares, multiple linear regression models, and the intelligent consensus predictive tool of the Small Dataset Modeler software based on Periodic table and physicochemical descriptors. It was shown that an increase in the number of nanoparticles, formation of a metal cation, solubility, and electronegativity led to the increment of cytotoxicity [124].

Winkler et al., employed Bayesian NN using sparse machine learning methods to develop proof-of-concept models using molecular descriptors and studied the biological effects of various nanoparticles: (a) iron oxide decorated with 108 different molecules regarding their toxicity to five cell lines, (b) 52 nanoparticles varying in composition and functionalized using surface attachments or coating with zeta potential, relativities, and size used as main descriptors for their biological effects on four cell lines, using 4 different concentrations, and (c) GNPs, which were modified using different small molecules and studied about their ability for non- and specific binding to biological endpoints. The developed model included a Bayesian regularized feed-forward NN with three layers, using either a Gaussian or Laplacian prior, combined with non-linear feature selection method such as multi-linear regression optimized by an expectation minimization algorithm with a sparse (Laplacian) prior to maximize the complexity level and the predictive ability, sparsity, and weight pruning of the model. Such algorithms have the advantage of being very fast in handling high-throughput data, and can support the analysis of synthesis, characterization, and testing related to nanomaterials and their biological effects, which have the potential to support policymaking in the near future. Thus, these predictive models can be utilized to assist regulators to ensure a safe working environment, as well as protect the environment and the eco-systems from the nanomaterials acute effects without restricting innovation in nanomaterials development and their commercial exploitation. In their research, it was shown that surface charge or zeta potential have a weak correspondence to cell apoptosis, which is regarded to relate to an adsorption mechanism of protein corona in serum or plasma of the cell culture, which introduces a zeta potential in the region of −10 to −20 mV, and thus the individual zeta potential of the nanomaterials is not relevant for the models. In regard to smooth muscle apoptosis mechanism, the type of nanomaterials was shown to be more important, especially for nanoparticles consisted of transition metal oxides, where cellular cytotoxic, proinflammatory responses and their ability to generate oxygen radicals were the key descriptors. Another outcome of the study was that macrophage cells are not able to uptake nanoparticles, especially when the size does not exceed the 30 nm, and it is effective if the size exceeds the 300 nm. Finally, there was not clear connection of the surface chemistry of nanoparticles and the macrophage uptake [118].

Particle size is a widely accepted descriptor which influences various toxic or biological effects of Ag nanoparticles, along with other weaker features, such as shape, agglomeration/aggregation states, surface chemistry or colloidal stability. Cell uptake, damages to DNA, mitochondria and cell membrane, and ROS production have been identified as effects that are mainly size-dependent, which rises the importance of a standardized and widely accepted monitoring technique, which can provide high-throughput information about the statistical representation of size distribution. Current techniques suffer from the agglomeration state of nanoparticles in the case of DLS, while in TEM observation the nanoparticles might be affected by the test media [27].

In the study of Yu et al., a dose dependency on toxicity effects of metal oxide nanoparticles has been demonstrated via the interpretation of machine learning models, which showed that the hydrodynamic size of nanoparticles has a negative impact on cells viability when it exceeds the value of 260 nm, which was mainly evidenced in case of CuO and Mn_2_O_3_ nanoparticles. An overall higher cell toxicity was also observed in case of CoO and ZnO nanoparticles [49]. However, another major outcome of their study was the fact that descriptors demonstrated high correlation, which makes it difficult to identify the parameters that influence the toxicity by a horizontal overview of the nanomaterial parameters, thus limiting the transfer learning ability of the developed models.

Of course, someone can recognize that in such data driven representations both the sensitivity and the degree of correlation between parameters are dependent on the amount of data available. If a researcher utilizes the data from the studies presented above another parameter can show the stronger influence on nanomaterials activity, whereas strong correlated parameters may be less correlated when the context is more general. Thus, the scientific community should recognize when the conclusions of a research are domain specific or could be more horizontal, which is the case. The ultimate target should be to reduce the experimental effort, while in-silico methods enable to reduce uncertainty of the outcomes, and efficiently reduce the resources requirements for safety assessments which will influence the design and engineering phase of nanomaterials to realize win-win scenarios. In addition, machine learning should be used in a manner that does not cancel previous outcomes, norms, or rules, unless this is a breakthrough revelation.

In another study, machine learning classification algorithms and NN clustering were adopted to map the nanomaterial genotoxicity/mutagenicity and perform fast screening to identify the best candidates using a safe-by-design approach. Based on a set of selection rules it was shown that the algorithm could accurately predict the mutagenic character of NiO, In_2_O_3_, CuO, and WO_3_ nanomaterials based on Ames test using the covalent index classifier, while the Comet test rule, which includes also the impact of cation polarization power, was employed to predict genotoxicity [125]. On the contrary, it was revealed that generally the size of nanoparticles does not specifically affect their genotoxicity, and thus Sizochenko et al., in their study developed a model without such functionality, and neither the shape nor solvents were used as descriptors.

Another approach by Sizochenko et al., includes the use of SOMs to perform interspecies correlation analysis modelling in the nanoscale to predict both qualitatively and quantitatively nanomaterials cytotoxicity and different mechanisms were revealed. More specifically, they studied 184 nanomaterials including metal and silicon oxide nanoparticles, using 15 datasets about their toxicity to algae, protozoa, bacteria, and mammalian cell lines, and the nanomaterials were categorized in four classes based on their activity. Special effort was devoted to identifying periodic table descriptors, which correspond to nanoparticles charge, valence, electronegativity, ionic radius, and other relevant properties. Relevant developments in the specific field have been realized by utilizing a simplified representation of molecular structure, metal ligand binding descriptors, and liquid drop models, assisted by quantum chemical methods. Another paradigm for addressing these challenges has been identified in terms of descriptors such as sphericity, as derived by TEM or SMILES structural representations, which can flourish under the development of CNNs and deep learning models, which have the advantage to be unbiased by the human factor in the selection of suitable descriptors [45].

In the study of Ban et al., zeta and redox potential, as well as dissolution rate were shown to determine the extent of nanomaterials toxicity, whereas it was revealed that surface modification of nanomaterials was strongly correlated to the ability of protein corona formation [49]. Feng et al., gave another perspective about the toxicity of functionalized nanomaterials. In addition to the surface chemistry, the shape effects on nanomaterial inflammation effect were also studied, and shown to be dependent on the affinity in cells-nanomaterials interface. It was shown that is not about the nanomaterial itself, but the presence and the expression of inflammatory factors, such as bacterial endotoxins (i.e., lipopolysaccharides) or surfactants that ensure well-dispersibility and are used in formulations. For the case study of Au and Pt nanomaterials, such endotoxins have the ability to bind on the surface and interfere with the formation of biological coronas, which trigger the monocyte inflammatory reaction. In this case, inhibition of inflammatory response was achieved by using the purified forms of nanomaterials, which raises a major issue in establishing actual relation of nanomaterials and toxicity effects, as then the modelling can be significantly biased by the presence of contaminants when using experimental data, which is not the case in the vast majority of studies and may lead to false-positive conclusions about immune response or false-negative outcomes related to immunotherapy studies [37].

### 4.2. Machine Learning for Nanomaterials Applications in Biomedicine and Therapies

Another concern of nanomaterials is their applicability for medical use, which is limited by the authorization by administrative bodies, such as FDA [58]. Nanoparticles hold a lot of promise to serve improved cellular uptake and successfully target cancer cells in order to deliver biological factors that are designed to fight the specific malicious cells. To support these efforts dedicated nano-QSAR models have been developed in order to estimate the acute effects and efficiency of nanomaterials, to reduce the need for performing a high-cost investigation which is required due to the large design space. Several models have been reported concerning the toxicity of metals, their oxides, and nanoparticles, but still there is a gap in the range of cellular toxicity studies, e.g., regarding the effects on liposomes or micelles cytotoxicity, while other classes of nanomaterials such as polymeric nanoparticles have not been systematically studied. In other cases, the studies are limited by the computational verification of the strong correlation of cytotoxicity with nanomaterials features, to name a few, size, concentration, and charge. Moreover, in many research approaches, what is missing is the high-dimensionality and the amount of data used for constructing the datasets, which outshines an overall need for aggregation/collection of more data about cytotoxicity in order to develop machine learning models that demonstrate generalizability, reliability, and transfer learning capability, which is often limited by overfitting and class imbalance [58].

What is more interesting to study in case of nanomaterials is the toxicity when inserted into blood circulation; the binding of several proteins onto the surface during circulation may alter their properties and biological fate. Lazarovits et al., studied this specific case by sampling at different times and isolating the nanoparticles, and quantifying the attached proteins via mass spectrometry. These inputs can be used to train machine learning in order to predict the organ accumulation of nanoparticles, biological fate, and also the blood clearance. A DNN was developed by the authors, which has the competitive advantage over hierarchical clustering and principal component analysis to identify specific patterns on high-dimensional data even if variation in data is not high. The model was used to study the accumulation of nanoparticles in spleen and liver organs, while the predictive accuracy reached the 94% using a double-blinded study. There were evidenced numerous different combinations of nanoparticles and binding proteins, while also there are multiple chemical/biological inhibitors that can enhance their clearance from the blood, while the clearance patterns were found to be predictable and it was possible to reduce spleen and liver uptake by 70% and 50%, respectively. Moreover, other interactions of nanoparticles contain the interactions and alternations by organs, tissues, and cells, which result in nanoparticles fate as a mechanism that inhibits their toxicity, and results in minimized accumulation by the target site. Currently, the unpredictable lifecycle of nanoparticles, including multifactorial and multivariate changes, in the human body has limited their application in clinical nanomedicine. The authors used in their study the GNPs as the model nanomaterial since their synthesis is well controlled over size and surface chemistry, while another benefit is that ICP-MS characterization can effectively quantify the Au nanoparticles elementally and PEG coating can ensure their long half-life in circulation in the tested rats. The dataset consisted of multiple parameters, including 63,630 proteins’ label-free quantitative (LFQ) intensities, quantified Au in the blood over 24 h, gold content in liver and spleen, and 5 different Au nanoparticles sizes (8–80 nm). It was proved that the increment in nanoparticle size led to increased clearance rate, and subsequent uptake by liver and spleen. The deep learning model demonstrated over 90% accuracy in predicting the injected nanoparticle size, while the liver accumulation was predicted with 81–93% accuracy, in case of spleen uptake the same metric was in the range of 77–92.6%, the blood half-life of nanoparticles was predicted with 84–91% accuracy and for the biological fate 77–94%. However, a big challenge for therapeutic applications remains the prediction of nanoparticle lifecycle considering that the multitude of proteins adsorbed may be difficult to manipulate and the distribution patterns difficult to alter, which was proven by the assay of Serial Injection of Materials for Biodistribution Analysis (SIMBA). Thus, these challenges were addressed with supervised machine learning in their study to describe how the surface chemistry of nanoparticles dictates the interactions with tissues and cells in the body, and efficiently predicts the biodistribution and clearance patterns of engineered nanomaterials and contribute to the design of nanomaterials and relevant strategies for novel therapeutics development [126].

The in-silico design of an innovative therapy will contain the selection of nanoparticles based on the different capabilities to carry drugs to the target organ. In such a case, an evolutionary algorithm has been developed by Tsompanas et al., to provide a metameric representation. The algorithm demonstrated adaptivity to the genome length and the number of the combined nanoparticles to exploit the same fitness with the training dataset. The simulation indicated that therapeutic nanoparticles should be released at a high rate and early in the process to increase the effectiveness. It is proposed by the authors that the high cost of the simulation can be significantly reduced by the involvement of machine learning in the evolutionary optimization loop, which is the case in surrogate-assisted evolutionary algorithms [40].

Esmaeilzadeh-Gharehdaghi et al., in their study developed ANNs to identify the optimum parameter to accomplish the monodispersion of nanochitozan nanoparticles, which is essential to realize high in vivo/vitro performance, physical stability, carrier capacity, release profiles, and enhanced biodistribution of active ingredient and degradation of biopolymer nanoparticles, as well as to obtain a predictable clearance profile for application in nanomedicine. The polydispersity index (PDI) of chitosan nanodispersion (107–710 nm) was modeled in a solution medium using four uncorrelated parameters in order to receive feedback from the interactions between different experimental factors, and more specifically the concentration of chitosan, pH, amplitude, and duration of sonication showing a reverse relationship. Using the developed model, it was possible to obtain the optimum parameters for achieving monodispersity with PDI equal to 0.2, while concentration parameter was considered to be the parameter that affected the PDI value at most [127].

In another application of machine learning for studying the targeted activity of nanomaterials, two comprehensive datasets are the ChEMBL and the DVRNs to develop a general-purpose model developing DVRNs for cancer co-therapy by-design. In the first dataset, a considerable number of assay conditions is included, with features that describe biological activity, types of proteins, assay strains and organisms, while the second describes 25 nanosystems with 16 features about biological activity, assay cells, and raw nanomaterials amongst them. Santana et al., used moving average operators to quantify deviations of the input variables, and multiplicative PT operators to reduce dimensionality and perform data fusion, while linear discriminant analysis was also used to find the optimum machine learning model. It was shown that the derived model was capable of addressing the accuracy, sensitivity and specificity requirement, as indicated by the validation by more than 130,000 cases (DVRNs vs. ChEMBL data pairs), and the overall model accuracy in the range of 83–88% [25]. A downfall of the developed model is the requirement for the exploration of a wide space of moving average parameters, which can be performed in a more efficient manner by using feature selection methods, as well as another restricting feature is the requirement to include 21 variables, which is connected to the boundary conditions of the biological assay of nanoparticle drug release system, of the vitamin and its synthesis parameters. For the second case it seems that a feature selection method may be prone to information loss, so that alternative approaches such as perturbation-theory operators (PTOs) may give a better alternative for efficient reduction of dimensions according to the relevant metrics (Chi square, p-level), with an ultimate target to predict biological activity (inhibition %) and pharmaceutical function (cumulative release %). The training dataset consisted of almost 90,000 observations, including almost 6700 observations with desirable biological effects, while the rest were non-desirable.

Another special case of nanomaterials is DNA, which is not actually considered as a biological material in the nanotechnology field. Engineering functional sequences of DNA can be key to produce nanomaterials with functional properties, which is the case for example in antigen-specific monoclonal antibodies (mAb) development assisted by generative machine learning computational design can find application in cancer treatment and autoimmune therapies development. The bottleneck in this case is the ability to experimentally verify the utility of the possible antibody sequences that can be combined, considering the main design parameters, such as epitope, paratope, affinity, and developability. A possible solution was proposed by Akbar et al., with the development of a simulation network to describe lattice-based antibody-antigen binding, which integrates a representative framework of parameters for physiological antibody binding. Machine learning can be used in this case to generate virtually 3D-antibody-antigen structures and predict their functionality. More specifically, 1D sequential data of 70,000,000 structures (higher amount of data by three orders of magnitude compared to SoA were used to feed the transfer learning model which reflects proper biological complexity in order to design the conformational (3D) epitope-specific antibodies, while the deep learning computer mAb design at high-throughput was validated by experiments for its ability to accelerate antibody discovery. A key functionality of in silico generative frameworks was that since training is completed, the model can be utilized as on-demand as a large-scale tool for the development of virtual representations of antigen-specific and immune receptor sequences [128].

Machine learning has also found application to the vaccine design field, and more specifically Munteanu et al., used predictive modelling for proteomes to be used as new B-cell epitopes, whose activity is a function of the queried peptide sequence, which in this case is used as a descriptor. Seven models were developed using 709,100 instances for training (IEDB database http://www.iedb.org, accessed on 31 July 2022), which contain data for 505 adjuvant additives, 28 experimental methods, 15 types of in vivo processes, 323 host and 1448 epitope organisms, and 83,683 peptides sequences, which included 10 features about the reference and query peptide sequences. It was shown that the random forest model better described the design problem using only 50 trees even for the classic QSAR model, as it was evaluated by the metric area under the receiver operating characteristics (AUROC) which resulted in a score of 0.981 using five-fold cross-validation to unknown data. The sensitivity analysis showed that the reference epitope activity and perturbation of the Shannon entropies were the more important descriptors. The model introduced by the authors is expected to have an impact in the in silico screening of peptides, and contribute to the field of vaccine design by taking advantage of the epitope prediction and the established structure–activity relationships [129].

### 4.3. Machine Learning Modelling of Environmental Effects

Machine learning is expected to be an emerging solution in the shortage of resources and support the supply chain by contributing to enhanced management of energy, resources, and time toward waste minimization [53]. This is a major gain for the manufacturing processes and industrial resilience, since machine learning can be used in Graphic User Interfaces (GUIs) and support the real-time adaptation of the parameters or predict the expected outcome early in the process without any adaptation to prevent quality issues. What is more, it is possible to identify processing step shortcuts and provide diagnostic insights about flaws in materials behavior or processing units. Another important functionality is to support the data-driven optimization, which can be used to adapt the process parameters towards self-optimizing reinforcement learning control systems and a tailored outcome, i.e., control and tune physical properties (optical, magnetic, electronic) on the nanoscale and/or predict the unknown levels of contaminants which affect the environmental toxicological profile based on previous experience, in regard to material properties and service life expectations.

Energy is another field that is moving towards the protection of the environment with the innovations in the development of energy conversion technologies using nanomaterials assisted by in silico methods. For instance, in the case of silicon based solar cells, the current limitation is that the band gap cannot convert quantitatively all the photons received onto the surface to energy. Two main parameters affect the energy conversion. Specifically, the bandgap of solar cells should be equal to 1.34 eV and the wavelength of the solar radiation received should vary from 400 to 700 nm. Out of this range, the photons cannot be captured due to the Shockely queisser limit, with their energy being lower than the band gap energy of the solar cell. Possible strategies to overcome this limit include altering the solar spectrum over 700 nm in order to increase the number of photons received or to use nanomaterials which are able to convert a larger portion of solar energy to current due to their band gap. The current limitation of nanomaterials is their high cost, but the silicon solar cell cost is still higher, and the cost–benefit balance can favour the adoption of nanomaterials in this case. Two promising candidates for improving conversion efficiency are silicon nanoholes/nanowires (SINH) and Graphene and its oxide form. For Si nanowires the large surface area to volume ratio is an enabling feature to improve light harvesting, while higher charge collection is facilitated by the formation of a core-sheath p-n junction. In the case of graphene, the advantage of the high conductivity is not combined with capacity for energy storage, while graphene, which is considered as a supercapacitor, is expected to be a more suitable candidate for solar energy conversion due to the better charge holding capacity and its ability to serve long time solar energy storage. Another strategic benefit is the substitution of the potential toxic constituents of batteries and released chemicals, which are not re-usable in the end-of-life, while for example graphene is considered to be biodegradable in nature without introducing any hazard to the environment. In this scope, to support the solar cell development with increased efficiency, machine learning was used to estimate the solar energy harvesting for a period of three years using the Sun path calculator application to collect the required data, which were 13,140 in number. A DNN was developed, using Adam optimizer and 20 epochs in total, to deliver a model with almost 93% accuracy in both training and validation datasets. It was shown that the nanomaterials can introduce actual improvements in the maximum solar energy that can be harvested. More specifically, it was revealed that graphene can lead to a 40% increment of harvesting capacity in comparison with silicon solar cells, based on the sun path tracking with machine learning, which is expected to create new opportunities in the smart farming of renewable and eco-friendly energy alternatives [130].

The modelling of environmental impact of nanomaterials by the utilization of sensors (including biosensors) is another important sector in order to acquire high-throughput and real-time data and feed machine learning algorithms and perform meta-analysis. One relevant example regards self-sustaining wireless sensor networks (WSNs), which are used for mining of data in order to supervise the personal health and environment monitoring. The current sensor technology though is limited to the monitoring of physical properties, which correspond to pressure, temperature, salinity, conductivity, moisture, movement/vibration or light illumination, while there is an emerging need to monitor also chemical descriptors. The large scale deployment of environmental monitoring has been evidenced for instance by the wireless sensor network GreenOrbs infrastructure (5000 sensors), which has been and can be used for surveillance of marine environment, water status, and greenhouse monitoring among others. In this field, wireless body sensor network (WBSN), including wearable devices, can find promising deployment via the mobile devices using the IoT to keep track for example of human health or even the environmental status by a data-driven feedback; still there is the need to provide the ability via wireless (bio-)sensors to track specific molecular information, by using interfaces with the blood, sweat, soil, or water, which have a tailored functionalized surface to act as a receptor suitable to study, i.e., the human and environmental condition after exposure to engineered nanomaterials [17].

Since nanomaterials have a bright potential for the demonstration of selectivity on emitted gases and chemicals, sensor development and integration with machine learning are quite promising to provide advances in the field of environmental protection. Thai in their work developed low-cost chemoresistive gas microsensors on a chip interface for producing on-chip grown tin oxide (SnO_2_) nanowires decorated with noble metals to produce four identical resistive nodes/sensors; one sensor was decorated with Pt nanoparticles and the second was decorated with Ag nanoparticles. The microsensors can support both portable and wearable gas sensing applications. In this case the authors integrated the sensors with a heater and positioned the nodes at different spots to acquire 4D data with the capability to identify five different gases, i.e., ethanol, acetone, hydrogen, hydrogen sulphide, and ammonia, and provide an estimation of the concentrations in a quantitative manner. A machine learning model was involved in order to support the handling of sensor data and automate the recognition process for the emitted gases. A support vector machine was used which was capable to classify (recognise the class of gases) the gases with 100% accuracy, while in the quantitative estimation of gas concentration, using the exposure limits of American Conference of Governmental Industrial Hygienists, it was shown that error varies between 8% and 28%. The quantification error was attributed by the authors to the noble metal which was used to decorate the SnO_2_ nanowires, so that the Pt decorated sensors showed an increased ability to accurate predict gases such as ammonia, hydrogen, and acetone, while Ag decoration did not favor the identification of specific gases. Another factor that affects the selectivity of gases for the identification by the sensor is the operation temperature, which is also the temperature of the sensor nodes. Thus, the performance of each sensor was evaluated in that study, as well as their combined detection ability using the support vector machine model [131].

Moreover, beyond the use of sensor-based systems, the establishment of machine learning can support the in-silico monitoring of the environmental impact of engineered nanomaterials solely based on in vivo and in vitro data. MODERN’s library was used in a meta-analysis to model the toxicity of 12 different engineered nanoparticles based on type of assay and the resulting EC_50_ values. The procedure contained the normalization of data using their mean value divided by the standard deviation (standardization) and the Euclidean distance was used to group the data, followed by a process to convert distances to weights. What was shown by unsupervised machine learning was that there is a connection of solubility of engineered metal nanoparticles and toxicity, with CuO and ZnO being the most toxic. On the other hand, only MgO nanoparticles were non-toxic/passive. Other metal nanoparticles, namely Co_3_O_4_, Al_2_O_3_, SiO_2_, Fe_3_O_4_, Mn_3_O_4_, and TiO_2_, demonstrated concentration dependent toxicity beyond 100 mg/L. Exception to this rule were Sb_2_O_3_ and MgO nanoparticles even at high concentrations of 100 mg/L. Moreover, it was revealed that ZnO, CuO, and Pb nanoparticles were toxic via a growth inhibition mechanism at very low concentrations in the scale of 1 mg/L [46]. The modelling of the environmental effects of engineered nanoparticles was based on the viability data of bacteria, algae, and protozoa; in that case it was shown that solubility is not a descriptor for environmental toxicity, but the electronic structure was directly correlated for the case of soluble and active CuO, ZnO, TiO_2_, and Fe_3_O_4_. Similarly, ecotoxicity effects were observed for other active engineered nanomaterials, and more specifically, Mn_3_O_4_, SiO_2_, Co_3_O_4_, and Al_2_O_3_. The third group of nanomaterials contained Sb_2_O_3_, MgO, and WO_3_ which were passive as indicated by the EC_50_ metric, with a value below 100 mg/L, which corresponds to a superphysiological dose exposure.

Another factor affecting toxicity is the surface transformation—protein binding—when nanomaterials are exposed to nature. Varsou et al., conducted studies using typical grades of water (Class I and Class V) to represent environments of natural habitats for freshwater zooplankton Daphnia magna microorganism at pH conditions at 7.6–7.8. They used a dataset containing five Ag nanomaterials with different coatings and six TiO_2_ nanomaterials with different capping agents. Several concentrations were studied at various media of Class I or V river water containing natural organic matter or HH (high hardness) Combo, including freshly dispersed and two years long dispersions. The effective concentrations (EC) were determined at an exposure duration of either 24 or 48 h. In their approach they determined the EC_40_ corresponding to the 40% of the initial population at 48 h, which was set as the toxicity threshold with high confidence. Thus, the dataset was consisted of a total of 353 observations (available in NanoPhasros database), which corresponded to 150 substances that were considered toxic, and the rest of them non-toxic. The numerical descriptors involved the DLS and TEM size, tested concentration, the zeta potential at pH ranging from 7.6 to 7.8, electrophoretic mobility, and medium conductivity, which were preprocessed via a Gaussian normalization method, and a sensitivity analysis followed using the Isalos Analytics Platform. The k nearest neighbors EnaloskNN Nanoinformatics tool was used to establish the machine learning model in order to study the ecotoxicity in silico, following ECHA’s recommended strategy for grouping of NM. k nearest neighbors was used in this case due to its suitability to establish accurate models for small datasets combining a low computational cost, while the predictive power was evaluated and validated in accordance with OECD criteria and a QSAR model. The established model was used to evaluate whether the environmental ageing and degree of agglomeration of nanomaterials is connected to the reduction of the environmental ecotoxicity. Amongst the aims was to identify if the ecotoxicity is inhibited in the media containing natural organic matter compared to the media containing only salt. It was shown that the agglomeration of the nanomaterials led to the inhibition of toxicity, which was affected by the size, the surface conductivity, and the surface charge. The subsequent reduction of the reactivity and dissolution potential in the case of soluble Ag nanomaterials, and the increased size were related to the reduction of bioavailability and uptake by the daphnids. Another mechanism that was evidenced was connected to the presence of natural organic matter which lead to the formation of an ecological corona, which reduced the toxicity of the dispersed nanomaterials [117].

Mikolajczyk et al., established (nano-)QSAR models to support the safe-by-design synthesis of hybrid nanoparticles, by using Decision Trees, and multiple linear regression models. In this scope, a GA was combined with multiple linear regression using QSARINS software to generate the most relevant variables with no correlation in order to be used as molecular descriptors. In case of decision tree algorithm, a decision stump technique was used, which was characterized by a single attribute, and a regression meta-classifier to improve the performance. The decision tree algorithm was developed in the Weka environment and its incorporation into the KNIME Analytics Platform. Beyond computational modelling, experimental validation was performed to improve TiO_2_ nanomaterials photocatalytic properties and reduce environmental toxicity, including several surface modifications approaches with noble Au/Ag/Pd/Pt metals. It was revealed that photocatalytic properties and cytotoxicity against eukaryotic cells are related in case of multicomponent TiO_2_-based hybrids. More specifically the parameters that influence damages to cells are the solubility, the composition in Au/Ag/Pd/Pt metals, the electronic properties such as conduction band energy level, and band gap, and the photocatalytic activity [132].

What is more, machine learning can play an important role in the hazard assessment of chemicals that are exposed to the environment, which often demonstrate commonalities regarding the descriptors that determine the ecotoxicity of nanomaterials. Hou et al., in their study used the physicochemical properties (mined by OPERA database—available at EPA’s CompTox Chemistry Dashboard) of chemicals in USEtox version 2.11, as well as their toxicity mechanisms to establish a data-driven model for the identifying their ecotoxicity in terms of HC_50_ metric, which indicates the 50% reduction of the initial population of studied microorganisms. This is a rather complex task, due to the biological, physical, and chemical processes that describe how chemicals interact and transform in environmental ecosystems. The common room of developments in the field can be correlated to the ecotoxicological impact of nanomaterials, since many nanomaterials can be used to carry chemicals towards several applications, such as self-healing and nanomedicine or chemicals can be present in the form of contaminants. The validation of the machine learning models was performed by involving ECOSAR and other QSAR models, while three models were utilized to select the best performing model in comparison to principal component, partial least squares, and ordinary least squares regression methods. The dataset contained data regarding 27 inorganic metals and 3077 organic chemicals, out of which 283 HC_50_ values corresponded to extrapolation by the known acute EC_50_ values, and 2307 contained actual both HC_50_ values and physicochemical properties; amongst them Henry’s law constant, boiling point, biodegradation half-life, bioconcentration factor, atmospheric hydroxylation rate, molecular weight, water solubility, vapor pressure, soil adsorption coefficient, octanol-water and octanol/air partition coefficient, fish biotransformation half-life. The authors demonstrated that the random forest model was more efficient to predict the ecotoxicity characterization factors and HC_50_ values compared to ECOSAR and the machine learning regression methods. It was shown that hydrophobicity and water solubility were the descriptors that determined the chemicals effect by an ecotoxicological endpoint, which is the case also for nanomaterials, which provided an efficient data-driven methodology, which can also guide the life cycle analysis of chemicals end of life environmental impact and is promising to enable broader applications [133].

Another green application of nanomaterials and machine learning is in the development of gas removal technologies, e.g., for the management of NO waste gas, in order to conform with the emission requirements for diesel and gasoline engines, as well as the development of catalytic technologies which do not require reducing agents for the decomposition of NO in automotive sector. The challenge in current metal catalyst technology is the poisoning due to the oxygen dissociation and their strong binding energy with oxygen. In the past this issue was addressed by using alloy catalysts including Au and Rh elements, which enhance the oxygen desorption. In this case, machine learning was used to correlate the binding energies with nanoparticles of atoms and molecules, as well as to predict the formation energies of nanoparticles and assist the decomposition reaction kinetics analysis, in order to extract useful information about the composition, size, and surface segregations to catalytic active, as well as individual features of the active sites present in the catalyst structures. The machine learning approach demonstrates benefits over the conventional Sabatier analysis. The latter requires many approximations to predict the catalyst efficiency. Actually, in their study, machine learning was combined with DFT simulation using the smooth overlap atomic position, which give information about the similarity of two local environments of the surrounding molecules/surfaces/solids in order to predict the energies of relaxed surfaces and reaction intermediates by using as input their unrelaxed atomic geometries, and multi-dimensional descriptors. A Bayesian linear regressor was used to provide the input parameters for DFT, which was capable of predicting binding energy of surfaces with the reaction intermediates [134].

In a similar work, structure activity relationships were established using a combination of DFT with ANNs to study the oxidation of GNP clusters in the presence of CO in an efficient manner by reducing the computational cost. With DFT the optimum structures of GNP clusters were defined, including anionic, neutral, and cationic structures, as well as structural descriptors (electron affinity, ionization potential, HOMO-LUMO gap, binding energy) and adsorption energies of O_2_ and CO, while the input parameters (defined by the user) were the coordination number of the adsorption site, the presence of unpaired electrons, the charge and the size of the cluster. The ANNs were used to correlate the structural descriptors with the aforementioned user defined parameters and the adsorption energy, which was shown to have similar capabilities to DFT. What was shown is that the adsorption strength of CO was higher for the cationic structures, which is related to the electron donating nature of CO, while size mattered only for cationic clusters and demonstrated a decreasing trend [135]. The latter is an expected outcome as in theory the charge is diluted by increasing the number of cluster atoms.

In another study, “atom types” were used as features of the dataset to feed the machine learning models in order to accurately describe the chemical character of materials and their ability to capture toxic waste gases, which was shown to be promising in regard to exceeding the performance of other models where the building blocks of the materials are used or the chemical character is not considered at all. By using the “atom types” approach, it was shown that significant improvements were achieved using smaller training sets by an order of magnitude, whereas compared to the building blocks approach it provides a strategic advantage of generalizability—the number of atom types is particular, while building block combinations could be unlimited—for deployment in studies of other kind of materials. This was specifically demonstrated by Fanourgakis et al., in the case of COF and MOF nanomaterials. The validation of the actual improvement in modelling performance was realized by using the database data of 137,953 structures and their atomic coordinates, that were formed by bringing together the MOFs building blocks, which include 16 functional groups, 30 organic linkers, and five different metallic corners, while the atom types were identified for almost 78,000 structures. Machine learning was established using a random forest model using the scikit-learn module for the identification of nanomaterial gas adsorption capacities based on capturing eco-pollutant gases—specifically, methane and carbon dioxide—and various thermodynamic conditions, as it has been proved efficient in regard to computational cost in past studies of MOFs compared to molecular simulations [136].

Another sector where nanomaterials can have a critical impact in the scope of environmental and human well-being is the automotive industry. Due to their catalytic properties, a reduction can be realized in regard to harmful emissions and the improvement of the conversion yield of waste gases to substances that do not have short- or long-term effects on health, such as nitrogen, water, and carbon dioxide. In this scope, current catalysts have shown limitations regarding their applicability, which are summarized as the conversion performance, including how performance may be decreased during lifetime of a vehicle, while also there is a temperature dependency. Machine learning can address these needs by the in-silico design of structures that have the potential to outperform current catalyst technology. So far, several efforts have focused on screening new catalyst candidates, on augmentation of quantum mechanics about the catalytic functionalities, and on prediction the properties of catalysts using dedicated descriptors. Whitehead in their study approached the problem by the scope of the process-property relationship, in order to identify the optimum parameters for the synthesis of the catalyst, including composition information of the formulations. The machine learning model was developed to predict 16 different catalytic properties which were verified by laboratory testing. To support decision making, Alchemite software was adopted in order to quantify the uncertainty of the predictions and establish a proper recommendation system for the next experiment. The training dataset was consisted of 551 catalysts, along with their information, while 10% of the data were excluded to be used as the testing dataset. A Bayesian tree of Parzen estimators model was used, and predictions were optimized using hyperparameter tuning and crossvalidation. With this study, the potential impact of machine learning to solve real-life problems where catalysts are needed was shown, as well as potential impact beyond the automotive field, e.g., in the design of metal alloys, drug carriers, or batteries, while it is possible to handle the sparse real-world experimental data, and support the full cycle of formulation development, including handling and establishing virtual representations of process–structure–property relationships for accelerated catalyst design [137].

## 5. Mining and Accessibility of Experimental Research to Enrich the Knowledge Base and Conduct Meta-Analysis

Machine learning has found many applications in identifying hidden patterns for the process design of nanomaterials, high-throughput screening in silico with the support of characterization data, as well in the design of safe bio-nanomaterials towards the discovery of novel therapies. Especially, predictive efficiency was shown in most cases not to be dependent on the algorithm complexity, but on the amount of data, their quality, and organization. Data mining is a subsection of AI and is responsible to browse and deliver data from multiple sources, including online, published articles, databases, images to enable the knowledge discovery. This technique enables the transformation of vast amounts of (un-)structured data into useful knowledge to address the modern needs of engineering community [9]. What is more interesting is that data mining in the fields of (nano-)materials science and engineering is considered as an extension of (Nano-)informatics [138].

The ultimate objective is to achieve a fully automated design of protocols in the algorithmic pipelines in order to automate the creation of workflows based on the nanomaterial targeted application. Of special importance is the development of cloud-based platforms that will enable the evolution, e.g., of wet-lab facilities for establishing an automated feedback system of experimental data (closed-loop laboratories) to allow the remote process design and monitoring with the capability for online and real-time adaptivity of manufacturing process, as well as the automation of digitalized data-driven optimization procedures [56].

Moreover, considering that these high-level ambitions of the industry transition in the following years are challenging the limits and the capacity for innovation, big data and machine learning approaches have a great potential to transform ambition to actual progress due to their generality and computational efficiency. In order to enable excellence at the industrial level, the establishment of large infrastructures with high computational capacity are required to enable handling of big data, as well as actions are required to support their usability and interoperability by establishing protocols for gathering and data curation. This is of high importance for materials science and nanotechnology, since access to large databases with structured characterization data and computational facilities is required to establish accurate predictive models. However, this is not limiting the utility of machine learning to use knowledge included in small datasets, whereas another advantage of machine learning is the computational efficiency in regards to resources and time. The actual limitation of machine learning is connected to fundamental conceptual difficulties regarding what machine learning can achieve in principle [17].

Another consideration concerns the available tools and characterization methods to generate high-throughput data, which is quite important to maximize the potential impact of machine intelligence. Still, domain experts need to supervise the experimental data in order to ensure that data are consistent with theory and knowledge acquired to date, in order to establish the fundamentals regarding experimental/simulation data annotation, validation, and possibly rejection to reduce bias introduced due to the low quality or due to the fact that new data are placed beyond the information space used for training the machines intelligence.

Data pre-processing is often a challenging task and requires a holistic understanding of how the algorithms function in order for machine learning to establish meaningful data representations. The following step of training the machine learning models is also an important step since the hyperparameters tuning are an individualized task for each algorithm. This is often quite tricky since the algorithms are required to perform accurately beyond the knowledge base, e.g., on unseen data, known as transfer learning, as well as to enrich the knowledge base, i.e., reinforcement learning, in order to take over challenging regression/classification tasks, e.g., as in the case of process and quality monitoring [17].

In the era of information, machine learning methods can be used to mine text and extract useful data features and learning rules using a large variety of algorithms in order to provide decision-making efficiency and accomplish real world specific tasks. Academic science and publications are flourishing, and it is well understood that the amount of data cannot be solely handled by human. In addition, search engines provide the ability to mine data based on limited number of keywords, and often it is a time-consuming task to browse relevant and usable information. A main priority is to utilize data from literature and store them in a way to be interoperable for all the researchers, and moreover to use the data already stored in databases, is to be able to trace the researchers workflow and satisfy that data provide the information without introducing bias into the conclusions of any research. Moreover, the collection of data, pre-processing, and dataset management/cleaning/formatting are essential processes for enabling datasets with different format to be utilized in extracting the desired information to enrich the knowledge base for training. A major concern of the research community, with all these years of technology developments and pushing for innovation, is the publication of unsuccessful research, in order to (i) ensure that the human effort and time are not spend without any outcome or result, (ii) to provide benefit to other researchers by saving time on what has been already performed, while (iii) giving answers when research is inconclusive. Machine learning can successfully address this concern, and utilize data that correspond to both successful and failed experiments, that can be prminent for the optimization tasks using machine learning, as well as guide the exploration in knowledge space. The knowledge space depends on data accessibility. The European Commission directives in Europe require that data are as open as possible and shared (as closed as possible) within the interconnected users via the web [60]. A major initiative towards this direction is the European Horizon 2020 Open Research Data Pilot.

Data that will be accessed and used for the development of algorithms that serve the materials identification, discovery, and optimization of crystallographic, optical, mechanical properties should be prepared following a specific protocol to nullify the potential errors. Often, data contain infinite/null/missing values or even different descriptor columns, so that the datasets should be harmonized regarding the contained columns and their identity [57,125], while also to exclude the rows which contain those problematic data cells or provide an estimated value, which usually corresponds to the mean value of two values in the neighboring region or use the most frequent values or other values based on similarity with other data [126]. Moreover, based on the principle used by the instrument to log the characterization data, it is possible to identify the abnormal values, which negatively impact any predictive model and the management of dummy values is critical when mining third party data. Other data pre-processing is connected to data sampling, and normalization/standardization, which are used in developing representative, reproducible, and accurate predictive models, ensuring that the values of data are not affected by the order of magnitude. One consideration is to use a normalization function that is reversible, in order to be able to revisit the original data and evaluate if the predictive models provide realistic predictions.

Ming Wang et al., introduced an automated feature engineering for the nanomaterial discovery. Automated feature engineering is another route for pre-processing of mined data, which extracts key parameters as independent descriptors, which are connected to specific structure and properties of nanomaterials. A common approach is to use deep learning algorithms and predict a desired output for discovery/design purposes, which is a costly process in regard to computational power and time. By using specified features of the dataset, it is possible to minimize the required costs and the domain knowledge for the establishment of a data-driven model with superior computational efficiency and experimental performance. Thus, feature engineering is essential when accessing large databases, which can tackle the need to access “Big Data” in the scale of ten thousand of data to build representative and descriptive models with transfer learning functionality [12], while the common case is to adopt transparent and abstract descriptors to ensure results accuracy.

Thus, text mining can overcome this shortcoming and increase visibility of new knowledge by the development of a systematic review process of tens of thousands full-text manuscripts and reduce the number of articles that should be manually reviewed by the scientists. For instance, it is feasible to automate the extraction of useful data from the published literature about polymer (natural/synthetic), organic/inorganic, carbon based, ceramic, metallic, and semi-metallic/semi-conducting nanomaterials at once in order to be used for meta-analysis, as shown by Li et al. Amongst the most popular methods for text mining are text clustering and classification, information extraction, and association rule mining. Thus, data mining can be a frontier in fields, such as the medical application of nanomaterials, in order to reduce their toxicity by design and effectively control their physical and chemical properties to the benefit of society [139].

Using mining to obtain data from published studies can provide new insights for the toxicity of nanomaterials that were not targeted in original publications and correlate some key attributes amongst the reported parameters with the toxicity endpoints. Meta-analysis using machine learning can reveal a data-driven trend also in cases that studies demonstrate conflict regarding the outcomes and resolve them by using the evidence by a structured summary of the studies, and even provide additional insights where the outcomes were inconclusive. For instance, in the mining of 17 publications about CNTs and their effect on pulmonary toxicity, 136 types of structures indicated that aggregate size, diameter, length, and metallic impurities are the most important descriptors. In another case regarding quantum dots, it was shown that toxicity was connected beyond the structural factors also to exposure indicators and cell lines such as exposure time, assay type, diameter, surface modification, shell and ligand [122].

For text meta-analysis and automated labelling of research datasets Naïve Bayes and k-means algorithm has been used. A demonstration of research highlights that trend prediction has been realized amongst 350,000 scientific articles published in 22 journals in the period of 2000–2017, showing that polymer nanomaterials are the most researched nanostructures, but the interest has been decreasing in recent years, and now the focus is on metal and carbon-based nanomaterials, which is predicted to be studied comprehensively in regards to biomedical effects in the next years [139]. In this direction, machine learning showcases the combination of text mining and computer vision as a promising strategy in order to mine hundreds of thousands of data reported in literature to link immune response to the nanomaterials and reveal the underlying mechanism of cell interaction [37].

Text mining has been realized also for the design of nanocarriers, by extracting 652 data for protein corona formation on various nanomaterials, including 21 qualitative and quantitative factors, amongst them structural, physicochemical, and experimental descriptors. The types of nanomaterials varied, and more specifically liposomal, non-metallic, metallic, and carbon-based nanomaterials were studied, including surface functionalized or surface charged, and pristine for modelling their interaction with 178 types of selected proteins and 73 corona components. The establishment of a precise model was quite challenging based on the diversity of the toxicity effects based on only limited factors considering the numerous variables that determine the nanomaterials behavior and cell response upon exposure. The sensitivity analysis showed that nine independent descriptors were important to the resulting toxicity profile, namely centrifugation speed, time, and temperature, incubation time, nanomaterial and plasma concentration, zeta potential, incubation plasma source size, and surface modification. Prediction accuracy for a random forest model reached a *R*^2^ metric higher than 0.8 regarding the prediction cell recognition of nanomaterials in both fetal bovine and human serum, and it was considered as a promising method to provide a platform for the design of nanocarriers with known biological responses and corona fingerprints [140].

In another study, knowledge discovery in the field of nanomaterials informatics was implemented with data mining algorithms to extract information for vapor-grown carbon nanofibers reinforced vinyl ester nanocomposites, and more specifically about the viscoelastic properties, which are described by materials properties of tan delta, storage modulus, and loss modulus. The input parameters contained the testing temperature, carbon nanofibers type, weight fraction, mixing method, and dispersing agent. SOMs showed that temperature was the most important descriptor, while the second most important was weight fraction [138]. It was shown that storage moduli drops with temperature increase, while beyond the glass transition temperature the effect was magnified by a decrease of several orders of magnitude [138].

Databases can be used also for knowledge discovery and support the systematic investigation of structure–property relations, since they provide a structured architecture which can increase the potential impact of machine learning. Two amongst the larger databases that have been open to public for years and used for mining, PubChem and SMILES, have made an impact on science by collecting data about chemicals and proteins. PubChem database provides multiple properties, such as structural, physicochemical, and bioactivities, including annotation, such as SMILES and chemical structures amongst them, which have already contributed to cheminformatics, medicinal chemistry, and drug discovery. The Protein Data Bank includes 3D structures of biomolecules, which have been actively involved in biology experimental and in silico research [8]. Another publicly available database, PubVINAS, contains annotated nanomaterials information suitable for modelling applications and to support rational nanomaterials design (http://www.pubvinas.com/, accessed on 31 July 2022). The database contains 705 unique nanomaterials, including GNPs, metal oxides nanoparticles, CNTs, DNA origami nanoparticles, and cyclic peptide nanotubes, including their structural descriptors, 1365 physicochemical and 2386 bioactivity features in PDB format. Yan et al., used these descriptors to develop property/bioactivity models with k-nearest neighbors and deep learning to predict three key factors determining the bioactivity profile, namely the cellular uptake, zeta potentials, and hydrophobicity of new nanomaterials from nanostructures [5].

Beyond PubVINAS, there are also other six databases (caNanoLab, eNanoMapper, NR, NKB, NBIK and NIL) for nanomaterials, containing structural characterization data. A common descriptor in all databases is the aspect ratio and shape, while except for the NIL database, the size (and distribution) and coating/shell information is available in all databases. Moreover, surface area information is available in some of them, while functional group descriptors are available only at NBIK, to support the understanding that the physicochemical properties are scattered amongst these seven databases. In addition, these databases contain biological properties derived from 23 common types of biological characterizations, experiments, and protocols for the nanomaterials contained, with emphasis on caNanoLab, eNanoMapper, and PubVINAS, which also contain risk assessment and safety evaluation of nanomaterials and derived products. Moreover, one consideration is regarding the functionality of the seven databases, what is missing is filtering functionality for NIL database, and searching functionality for PubVINAS, while another consideration is about the size of nanomaterials databases considering the amount of data generated in nanotechnology field. Annotation of nanomaterials may enhance the efforts for data collection and structuring in the future, while large nanomaterials databases are needed, such as PubChem, PDB, or EADB for instance, to facilitate rational nanomaterial design and the establishment of data-driven structure–activity relationships, and support industrial uptake and commercialization of nanomaterials, and safety, ethics, and regulation-compliance as well [55].

More databases have been established as well, to enable researchers to access real-world data for the analysis of 2D nanomaterials. Some indicative open-source databases, are the Harvard Clean Energy Project, the Open Quantum Materials Database and the Materials Project, ChemSpider, NanoHUB, Springer Materials, etc. For instance, mining of data from Materials Project Database led to development of models for the classification of 2D materials based on lattice constants as a descriptor with an accuracy of almost 89% amongst many 2D materials. In another case the International Crystallographic Structural Database was used for the selection of 2D materials based on covalent radii, structural gaps, packing ratio, and symmetry, as well as for the discovery of 92 new 2D structures, and the detection of monolayered 2D structures, which are stable using a topology-scaling algorithm [18].

What is the next step? An interesting strategy to enable visibility and maximize the impact of a database is the deployment of online application integrated with the databases and AI (as depicted in Figure 5), thus transforming databases to digital experience platforms and open innovation ecosystems. A paradigm is the dendPoint user-friendly application (http://biosig.unimelb.edu.au/dendpoint, accessed on 31 July 2022), which is a widely available in silico model, which utilized the physicochemical and scaffold structure properties of complex polymeric nanomaterials to predict intravenous pharmacokinetics. In that case, Kaminskas et al., manually curated a database including dendrimer biopharmaceutical behavior, and structural and chemical features of polymeric nanomaterials as well. This web application enables both the prediction and comparison of dendrimer key properties, such as dose and clearance in liver and urine, volume of distribution, and half-life, to facilitate the rapid exploration of literature derived properties, and to enable the visualization of dendrimer pharmacokinetic properties. This relational database and predictive method binding could be the catalyst for guiding the design and development of such structures by enhancing the scientific community efforts towards a systematic analysis of structure and the subsequent biological endpoints. The constant updating of such infrastructures is essential to reinforce research efforts to go beyond the SoA, while it can provide multiple functionalities when data become available and curated. In this case, a possible exposure scenario is that dendrimers may be delivered via non-intravenous routes, i.e., inhalation, which requires additional information concerning the physicochemical properties to accurately model the pharmacokinetic behavior, and successfully support progress in the field of nanomedicine [141].

In the future, such relational databases could be interactively enriched by the users, in such a manner that the user uploads and curates or the data are curated automatically by involving a proper protocol, with respect to the GDPR rules and user privacy. Moreover, the need for data standardization and proper documentation can maximize the potential uptake of research data and accelerate the progress in multiple nanotechnology fields beyond the nanomedicine, considering that often the main barrier in AI concerns data quality, amount, and structuring in the databases.

## 6. Prospects and Conclusions

To summarize, AI and nanoinformatics have burst out in the recent decade in many fields of nanomaterials application, and is now moving to inverse design and modelling of multi-structure/multi-property relationships. AI has already been used to tackle challenges in the design and discovery of novel nanomaterials, to optimize structure and performance by assisting process design and unbiased decision making, to ensure safer and eco-friendlier nanomaterials. With the aim to bridge the gap in fields of inconclusive research structure–property–activity–toxicity and descriptive biotoxicity profiles have been established, assisted by knowledge discovery from mining of the published scientific works as well as curated databases. The applications vary and include biomedicine, smart integrated biosensors, environmental science, such as gas separation, energy management composite devices, (wearable) electronics, automotive and aerospace, catalysis, and structural applications amongst them.

Furthermore, feature engineering was shown to be critical for the improvement of data-driven representation and has been a key strategy to reduce the bias and overfitting problem of the developed models. Sensitivity analysis enables to identify the descriptors that are strongly correlated, so that one can choose suitable descriptors in the design/engineering phase of the experiments and reduce the workload in regard to computation resources needed to model the process and synthesis of safe, circular, and greener by-design innovative nanomaterials [49]. A competitive benefit of deep learning models is that it is not necessary for the scientist to manually perform feature engineering, since the algorithm demonstrates self-adjustment efficiency and the ability to select suitable features independently in a continuous learning process. For example, when small datasets are available, linear regression model can find limited success since in the multi-dimensional field, known as “curse of dimensionality”, where the parameter relation is complex, and thus Bayesian/naïve Bayes methods can find success, due to their ability to readjust previously adjusted parameters. The decision tree and ensemble method are also other popular methods to handle small datasets, but still it has been shown that less sophisticated algorithms can overcome the performance if there is access to higher amount of data.

Additionally, evolutionary methods and data mining can be used standalone and bring innovation in multiple disciplines of nanotechnology and support the development of a new generation in nanomaterials high throughput synthesis, characterization, and data analysis. The role of evolutionary methods and data mining can be essential for the demanding online and real-time optimization tasks in a closed-loop laboratory and maximize the development time, reduce the resources needed by utilizing in silico modelling and inverse design procedures, and improve process adaptivity, production yield, and quality assurance of nanomaterials and their nanoreinforced composites.

Another topic which may bring a revolution in science and the research of nanomaterials and their activity profile is the combination of simulation (DFT, MD, Monte Carlo, CFD, quantum-chemical simulations) with AI, and more specifically surrogate models and evolutionary algorithms. Along with the data-driven nanomaterial representations, mathematic, chemistry, biology, and physics models can be combined to adequately model, e.g., complex long-range interatomic interactions, such as van der Waals, to predict reactivity or even the behavior profile by a bioactivity/biotoxicity end-point. The capacity of machine learning to perform rapid calculations can be vital to get the benefits of simulation with increased computational efficiency, considering that simulation is quite demanding due to factors such as equilibration time, millions of iterations, and solving (series) of complex equations. Thus, the efficient combination of predictive modelling will contribute to the formation of general design rules and contribute to a hierarchical assessment of in silico designed nanomaterials.

Another benefit by the development of machine learning and computer vision is to support nanocharacterization, considering the fact that in the new era of high-throughput and digital characterization the data amounts that are generated will no longer be able to be handled by the scientists. Thus, reliable algorithms that will preserve computing consumption and interdependence of data should be developed. The dependable AI will not be limited by the algorithm performance, and apart from the limit by the characterization method resolution, it will be possible to model uncertainty, take the maximum out of the characterization data, and enable the statistic representation of material behaviors and properties, ensuring the objective judgment by machine intelligence to enrich and not replace the domain knowledge. These advancements are expected to support the improvement and refinement of science in various fields, guide computations and experiments where there is an identified gap for science and/or industry, and trigger the advancement of nanotechnology and instrumentation as well.

Practical applications of AI tools reported in this summary of the current SoA are listed below:High-throughput research space exploration of nanomaterials options/candidates;Image segmentation/object detection for statistical analysis of nanomaterials shape/size/agglomeration state/defects detection;Objective and decentralized decision-making based on multi-dimensional datasets to improve generalizability and evidence-based conclusions (i.e., phase analysis, anomaly detection);Design of experiments via genetic algorithms, PSO;Fast calculation of input values for modelling, instead of using time-consuming simulation, especially where absolute accuracy is not limiting;Data mining of publicly available datasets to enrich the knowledge base and extrapolate predictive models with increased accuracy;Use of ensemble algorithms to improve predictive capabilities when limited information is accessible;Establishment of models to correlate the chemical structure and physical, chemical, and physicochemical properties with the activity and (eco-)toxicity profile of nanomaterials, utilizing known activity profiles of well-studied materials.

Currently, the emerging need for standardization of the produced data, including annotation, management, high content screening, and handling, has been highlighted by the scientific community. Especially, the individual features of the generated datasets, including possible small amount of data, sparsity, variability, class imbalance, bias, missing, or descriptors variance amongst datasets. Thus, protocols for the interoperability, collection, and curation of data are required in order to be stored in databases and to be useable to address the community needs. Moreover, standards regarding the assessment of data quality and relative paucity can fill the gap for handling non-curated data, which will enable meaningful data-driven representations with AI.

Another need for machine learning is to ensure the reproducibility of the predictive modelling process, considering that the barrier in previous simulation activities was about the lack of proper repeatability even when simulations were performed within the same research group. Thus, machine learning requires the dissemination of examples and sharing of representative data to support the proper documentation of the decisions made for the development of an algorithm and to reinforce research efforts to enable the domain knowledge validation by ensuring that different studies will not raise more questions than the topics/challenges answered.

What is more, the main bottleneck nowadays has been the access to large amounts of data, which need to be curated and structured in order to allow AI to mature in the forthcoming years and transform the available data to domain knowledge. Overcoming these barriers will allow to adapt a process both experimentally and in silico to discover and produce nanomaterials and address the needs of each application. Again, in the latter case the “curse of dimensionality” is a big challenge for the data science community.

Future progress in nanoinformatics will ensure that diverse scientific and socioeconomic information is integrated to identify on time the trends and needs for the society and maximize the commercialization benefits of technology. Governance over nanomaterials data can drive their commercialization in an efficient manner by identifying regulation and policy gaps on time and thus increasing the societal confidence and acceptance for the adoption and use of rational nanotechnology in every-day life applications. Finally, another need concerns data security, which can be the case in databases and with the online deployment of web applications, since the future is oriented in cloud computing and interconnection with databases. A lot of promise for data security is held by blockchain technology, which can provide complete supervision, as in the case of decentralized cryptocurrency projects.

## Figures and Tables

**Figure 1 nanomaterials-12-02646-f001:**
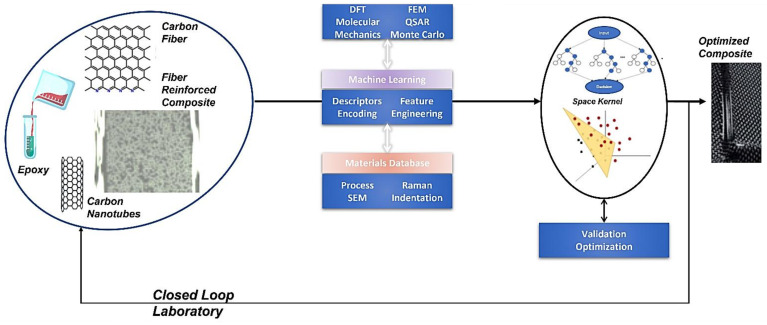
Schematic representation of a closed loop laboratory. Manufacturing, materials modelling and characterization can provide the data to correlate via data-driven AI empowered models the optimization of the next experiments in a closed-loop of information flow.

**Figure 2 nanomaterials-12-02646-f002:**
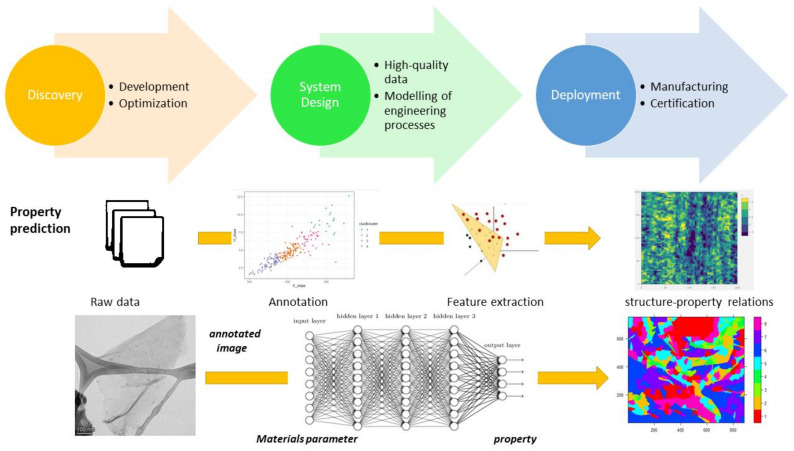
Illustration of machine learning contribution to nanotechnology and nanoinformatics field for nanomaterials design and selection based on data-driven analytics.

**Figure 3 nanomaterials-12-02646-f003:**
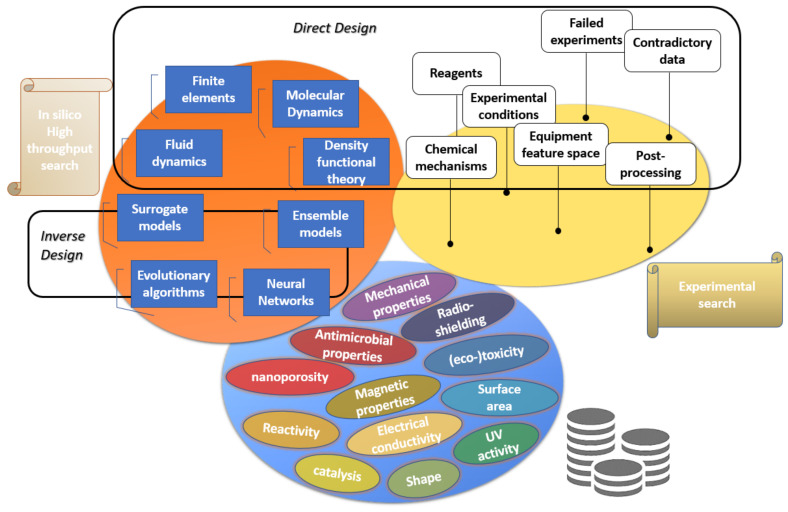
In silico high-throughput material screening and optimization to accelerate materials direct and inverse design.

**Figure 4 nanomaterials-12-02646-f004:**
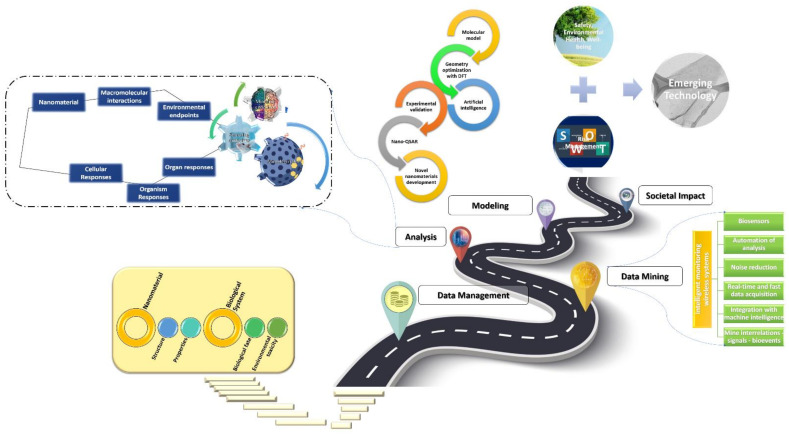
Predictive modelling applied for nanomaterials selection; impact in societal sciences, safety, and well-being.

**Figure 5 nanomaterials-12-02646-f005:**
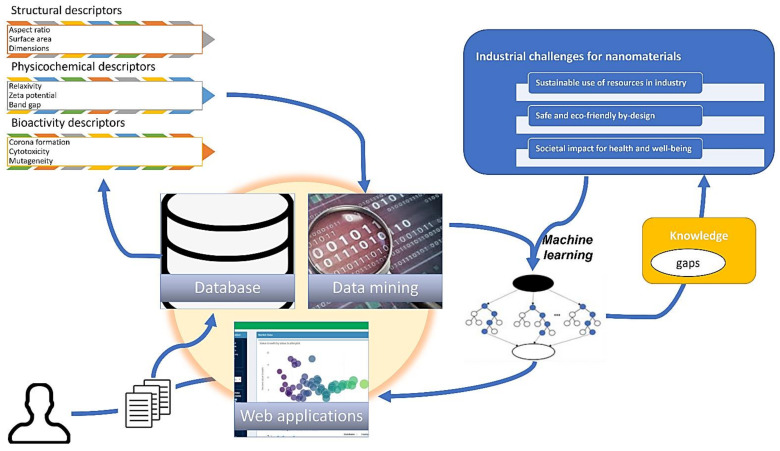
Database and knowledge discovery workflows to support innovation and circularity in Industry.

**Table 1 nanomaterials-12-02646-t001:** Landscape of recent reviews covering machine learning topics across different domains/fields of nanomanufacturing and machine learning applications.

Title	Scope	Refs
**Research and Development in Carbon Fibers and Advanced High-Performance Composites Supply Chain in Europe: A Roadmap for Challenges and the Industrial Uptake**	Novel materials and optimized processing Characterization and modelling Environmental and economical circularity	[4]
**Machine learning the ropes: principles, applications and directions in synthetic chemistry.**	Preprocessing, feature engineering, machine learning algorithms, databases Combinational simulation and machine learning for new material discovery High-throughput screening with machine learning Summary of recent progress in property prediction, material discovery, inverse design, corrosion detection	[12]
**Virtual metrology as an approach for product quality estimation in Industry 4.0: a systematic review and integrative conceptual framework**	Virtual metrology and Industrial application Selection criteria and overview of limitations	[13]
**Big data and machine learning for materials science**	Big data (including sensor based) and machine learning Materials discovery and quantum chemistry Trends, proposed future developments, practical limitations and cost of big data production by sensor systems	[17]
**EU US Roadmap Nanoinformatics 2030**	Data collection and curation Metadata and ontologies Nano(bio)informatics, machine learning and statistical modelling Impact, challenges, and milestones	[20]
**Materials discovery and design using machine learning**	Typical machine learning approaches and applications Limitations, routes for overcoming challenges and future orientations	[21]
**Advancing Biosensors with Machine Learning**	Chemometrics for different biosensors types and data Machine learning models advantages and limitations	[26]
**Knowledge gaps in immune response and immunotherapy involving nanomaterials: Databases and artificial intelligence for material design**	Nanomaterial technologies and immunotherapy Nanobioinformatics and databases	[37]
**Role of Artificial Intelligence and Machine Learning in Nanosafety**	Modelling of biological and environmental profiles of nanomaterials—computational nanosafety Summary of machine learning methods and descriptors Practical applications	[45]
**Toward computational and experimental characterisation for risk assessment of metal oxide nanoparticles**	Metal oxide nanoparticles Cytotoxicity and risk assessment In silico studies with QSAR models and machine learning	[47]
**Machine Learning for Advanced Additive Manufacturing**	Theoretical design expectations Practical manufacturing capabilities AI for AM	[50]
**Machine learning in additive manufacturing: State-of-the-art and perspectives**	Barriers for AM and inconsistent product quality Machine learning algorithms in design and in process for AM Data security	[51]
**NanoEHS beyond toxicity—focusing on biocorona**	Environmental health Safety of nanomaterials and laboratory factors affecting toxicity profile Biomolecular interactions at molecular level High-throughput screening with machine learning of materials properties towards greener design and statistical modelling	[52]
**A review of machine learning for the optimization of production processes**	Optimization in a wide range of processes and product quality Data generated in production, machine learning and constraints	[53]
**Machine learning for chemical discovery**	Curated datasets for chemical discovery Breakthroughs and challenges	[54]
**Nanomaterial Databases: Data Sources for Promoting Design and Risk Assessment of Nanomaterials**	Review of databases for nanomaterials and comparative analysis	[55]
**In silico design and automated learning to boost next-generation smart biomanufacturing**	Automated smart biomanufacturing (adaptive and rapid design) In silico tools, screening and prototyping in industry	[56]
**Practices and Trends of Machine Learning Application in Nanotoxicology**	Data pre-processing and machine learning model development Progress in nanoinformatics for nanotoxicology domains and in silico prediction	[57]
**A review of the applications of data mining and machine learning for the prediction of biomedical properties of nanoparticles**	Data mining and machine learning in nanomedicine—prediction of properties and activity Progress and challenges in nanoinformatics	[58]

## Data Availability

Data is contained within the article.

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
