# Peer review of "Digital Innovation Enabled Nanomaterial Manufacturing; Machine Learning Strategies and Green Perspectives"

_nanomaterials, 2022, doi:10.3390/nano12152646_

Round 1
Reviewer 1 Report
The manuscript entitled “Digital innovation enabled nanomaterial manufacturing; Machine learning strategies and green perspectives” could be considered for publication in Nanomaterials after minor revision. The following revisions are suggested:
1.For practical industrial applications, nanomaterial manufacturing must also demonstrate the stability. The stability of nanomaterial needs to be supplemented.
2.The abbreviation should be used at or after the second time, Ex. Artificial Intelligence (AI), carbon nanodots (CDs) and metal organic frameworks (MOFs).
3.The reduction in size and doping with other composites could affect in the performance of nanomaterial. The review should clearly indicate the development of nanocomposite in size.
4. Page 10, Line 439. The eco-friendly precursors need to be clearly stated.
Author Response
Reviewer 1
Comments and Suggestions for Authors
The manuscript entitled “Digital innovation enabled nanomaterial manufacturing; Machine learning strategies and green perspectives” could be considered for publication in Nanomaterials after minor revision. The following revisions are suggested:
1.For practical industrial applications, nanomaterial manufacturing must also demonstrate the stability. The stability of nanomaterial needs to be supplemented.
Answer: The authors have provided more information on the stability of nanomaterials, which has been evaluated using machine learning tools that enable the scalability of the agglomeration state detection utilizing microscopy images, as well as by the prediction of zeta potential of developed nanomaterials under the relevant sections.
Added:
Under the section 2: “In another study involving the physicochemical profile of TiO2 nanoparticles the zeta potential prediction was enabled by ANNs. Temperature, pH, ionic strength, and mass content of aqueous dispersions were studied with pH being the most influencing de-scriptor. Thus, the constructive model was regarded as a step beyond currently applied statistics and towards research space exploration on a wide range of conditions in sili-co, which due to technical and time constraints cannot be experimentally validated, but can offer a guidance on the next experiments. The impact in different industries, such as pigments and pharmaceuticals production, minerals processing, construction can be realized by tailorizing the synthesis or the manufacturing parameters towards an increased zeta potential value to favor the minimization of nanoparticles agglom-eration events and improve the sustainability of production when nanoparticles are used as reinforcements [59].”
Under the section 2.2: “Similarly, Ilett et al. introduced ilastik tool for object classification, which can be tailored to a wide range of parameters that can be used to deconvolute particles by microscopy images [68]. This tool includes a function of detection of different particles even in agglomeration state and corresponding quantification features which could be used to monitor nanomaterials stability and obtain more insights in the interpotential dynamics and the tendency to agglomerate, which is a critical aspect in nanomanu-facturing. Thus, machine learning was introduced to provide in that case the capacity to overcome the time-consumining manual process for identifying the agglomeration tendency and projected shape of agglomerates, as well as to overcome the limitations of DLS characterization, which as discussed above is prone to missing the agglomera-tion events since a circular shape of nanomaterials is assumed in all cases. In addition, it has been argued that DLS information on colloidal stability can be lim-ited since the suspensions studied with this method are stable colloids.”
2.The abbreviation should be used at or after the second time, Ex. Artificial Intelligence (AI), carbon nanodots (CDs) and metal organic frameworks (MOFs).
Answer: The authors have elaborated with the abbreviation use at first reference.
3.The reduction in size and doping with other composites could affect in the performance of nanomaterial. The review should clearly indicate the development of nanocomposite in size.
Answer: The authors would like to thank the reviewer for the suggestion. Indeed, the size is a key factor for nanocomposites development. The authors have been preparing another work to assess the nanocomposites development at present, as well as the challenges and next directions in manufacturing, and thus in this work the focus will remain on machine learning strategies by a computational point-of-view to avoid possible conflicts.
4.Page 10, Line 439. The eco-friendly precursors need to be clearly stated.
Answer: The authors revised the text to address the reviewer suggestion.
Added: “The bioinspired synthesis is characterized by the short reaction duration, the mild con-ditions at room temperature, and the use of eco-friendly precursor (sodium metasil-icate pentahydrate), …”
Reviewer 2 Report
The paper written by the following Authors: Georgios Konstantopoulos, Elias P. Koumoulos and Costas A. Charitidis entitled “Digital innovation enabled nanomaterial manufacturing; Machine learning strategies and green perspectives” presents an interesting study on machine learning application for nanomaterial manufacturing.
Although the paper is interesting, I have some major concerns:
Title
The title reflects the results presented here.
Abstract
The abstract is lacking the aim of the study.
Introduction
1. In the introduction part Authors should add some overall information about the initial and boundary conditions commonly applied.
2. Introduction part is lacking the aim of the study.
Conclusions
1. Limitation to the analysed topic should appear at the end of the manuscript.
2. Practical application should by highlighted.
Author Response
Reviewer 2
The paper written by the following Authors: Georgios Konstantopoulos, Elias P. Koumoulos and Costas A. Charitidis entitled “Digital innovation enabled nanomaterial manufacturing; Machine learning strategies and green perspectives” presents an interesting study on machine learning application for nanomaterial manufacturing.
Although the paper is interesting, I have some major concerns:
Title
The title reflects the results presented here.
Abstract
The abstract is lacking the aim of the study.
Answer: The abstract has been revised and improved, accordingly.
Added: “Machine learning has been an emerging scientific field serving the modern multidisciplinary needs in Materials Science and Manufacturing sector. Taxonomy and mapping of nanomaterial properties based on data analytics is going to ensure safe and green manufacturing with consciousness raised on effective resource management. The utilization of predictive modelling tools empowered with Artificial Intelligence has proposed novel paths in materials discovery and optimization, while can stimulate cutting-edge and data-driven design of a tailored behavioral profile of nanomaterials to serve the special needs of application environments. The previous knowledge of physics and mathematical representation of materials behavior, as well as the utilization of already generated testing data received specific attention by scientists; however, exploration of available information is not always manageable, and machine intelligence can efficiently (computational resources, time) meet this challenge via high-throughput multidimensional search exploration capabilities. More-over, modelling of bio-chemical interactions with environment and living organisms has been demonstrated to connect chemical structure with acute or tolerable effects upon exposure. Thus, in this review a summary of recent computational developments is aiming to cover excelling research, present challenges towards unbiased, decentralized and data-driven decision-making, in relation to increased impact in the field of Advanced Nanomaterials Manufacturing and Nanoinformatics, and indicate the steps required to realize rapid, safe and circular-by-design nanomaterials.
Introduction
- In the introduction part Authors should add some overall information about the initial and boundary conditions commonly applied.
Answer: The authors have thoroughly revised the Introduction section so that the strengths and the limitations of AI/Machine Learning tools are presented with clarity. If the comment of the reviewer corresponds to data handling/manipulation/prediction, the boundary conditions cannot be universal, and thus for the article robustness, the authors prefer not to add specific conditions but present the general context.
- Introduction part is lacking the aim of the study.
Answer: The authors revised the last paragraph of the introduction section to reflect the aims and objectives of the review more clearly.
Added: “This review focuses on the need to establish digitalized mapping between materi-als descriptors that arise from different characterization/simulation methods and re-late the chemical structure to nanomaterials options to favor the growth of dedicated properties profile and even to tune their production parameters to improve sustaina-bility. In section 2 the motivation is to introduce progress and challenges in discover-ing new materials by enabling shortcuts to rapid and safe-by-design routes, incorpo-rate experimental parameters in order to enhance decision-making and select the pro-cess parameters to produce nanomaterials to serve specific applications [40,43]. In ad-dition, in section 3 the improved capabilities for exploration of materials space and their corresponding digital representations are discussed and the key enabling steps to realize more flexibility in the production standards applied for selection of suitable nanomaterials, without the need to consider chemistry/physicochemistry or other complex theories in Materials Science, but using previous knowledge/experience doc-umented in corresponding datasets [40,45]. Another important mission of Machine In-telligence is in the field of the development of the safe and environmentally friendly nanomaterials, where in section 4 a threefold pattern in the prediction of their biologi-cal activity profile is presented based on data-driven representations, which could be prone to the availability of data and bias due to the applied procedures for sample preparation (contamination, purification), including biological systems containing human/algae cells, as well as in the bionanomaterial-cell interfaces. As presented in section 5, the foundation for this scope is data mining, management, and curation to enrich new or existing databases, as it is commonly accepted that the catalyst for AI is the accessibility to structured and Big Data [52,53]. Consequently, the existing methods can be supported by the computational theory that focuses on data, which holds a lot of promise to greatly reduce the computational and experimental pressure for innova-tion. At the end of this review the prospects of AI for the fields of Nanotechnology, Materials Science, and Nanoinformatics are summarized along with the conclusions of this study.”
Conclusions
- Limitation to the analysed topic should appear at the end of the manuscript.
Answer: The conclusions section has been restructured to summarize the main limitations at the end of the manuscript.
- Practical application should by highlighted.
Answer: In order to summarise practical applications of AI and machine learning, a list has been provided in the conclusions section.
Added: “Practical applications of AI tools reported in this summary of the current state-of-the-art are listed below:
- High-throughput research space exploration of nanomaterials options/candidates;
- Image segmentation/object detection for statistical analysis of nanomaterials shape/size/agglomeration state/defects detection;
- Objective and decentralized decision-making based on multi-dimensional datasets to improve generalizability and evidence-based conclusions (i.e. phase analysis, anomaly detection);
- Design of experiments via genetic algorithms, PSO;
- Fast calculation of input values for modelling, instead of using time-consuming simulation, especially where absolute accuracy is not limiting;
- Data mining of publicly available datasets to enrich the knowledge base and extrapolate predictive models with increased accuracy;
- Use of ensemble algorithms to improved predictive capabilities when limited information is accessible;
- Establishment of models to correlate the chemical structure and physical, chemical, and physicochemical properties with activity and (eco-)toxicity profile of nanomaterials, utilizing known activity profiles of well-studied materials. ”
Reviewer 3 Report
The authors are suggested to revise their manuscript based on the following comments:
Abstract should be more focussed and concised clealry presenting the need and the major topics covered in the review.
The introduction section is too long and contains many irrelevant details (far beyond the topic). The authors are advised to make it clearer and shorter with a major focus on the recently published articles (as well).
Please include a short summary (can be tabulated) of the prior reviews on the investigated topic.
At the end of the introduction section (or a separate section after the introduction), the aims and objectives of the review should be clearly described.
It is better to include a graphical abstract presenting the topics covered in the review for example names of the nanomaterials manufacturing technologies, the substrate materials, artifical intelligence/machine learning strategies ,and so on to cover the wide auience.
The short-comings and potential applications of the listed/presented AI/ML methodologies should be highlighted. In particular, the physical descriptors (explainable artificial intelligence perspective/role should be discussed). The authors are suggested to get help from the following articles; "Boiling Heat Transfer Evaluation in Nanoporous Surface Coatings", "Liquid-to-vapor phase change heat transfer evaluation and parameter sensitivity analysis of nanoporous surface coatings", "On the assessment of the mechanical properties of additively manufactured lattice structures" and the like.
The information presented in the discussion sections is better to be organized and in more readable/followable format.
The grammatical errors should be corrected.
Author Response
Reviewer 3
The authors are suggested to revise their manuscript based on the following comments:
- Abstract should be more focussed and concised clealry presenting the need and the major topics covered in the review.
Answer: The abstract has been revised and improved, according to reviewer recommendations.
Added: “Machine learning has been an emerging scientific field serving the modern multidisciplinary needs in Materials Science and Manufacturing sector. Taxonomy and mapping of nanomaterial properties based on data analytics is going to ensure safe and green manufacturing with consciousness raised on effective resource management. The utilization of predictive modelling tools empowered with Artificial Intelligence has proposed novel paths in materials discovery and optimization, while can stimulate cutting-edge and data-driven design of a tailored behavioral profile of nanomaterials to serve the special needs of application environments. The previous knowledge of physics and mathematical representation of materials behavior, as well as the utilization of already generated testing data received specific attention by scientists; however, exploration of available information is not always manageable, and machine intelligence can efficiently (computational resources, time) meet this challenge via high-throughput multidimensional search exploration capabilities. More-over, modelling of bio-chemical interactions with environment and living organisms has been demonstrated to connect chemical structure with acute or tolerable effects upon exposure. Thus, in this review a summary of recent computational developments is aiming to cover excelling research, present challenges towards unbiased, decentralized and data-driven decision-making, in relation to increased impact in the field of Advanced Nanomaterials Manufacturing and Nanoinformatics, and indicate the steps required to realize rapid, safe and circular-by-design nanomaterials.
- The introduction section is too long and contains many irrelevant details (far beyond the topic). The authors are advised to make it clearer and shorter with a major focus on the recently published articles (as well).
Answer: The original introduction section has been reduced by 20% (calculated based on lines numbering) and thoroughly revised, while 44 out of 58 references used in the introduction section (> 75 %) of the revised version correspond to publications within the last 5 years (2018-2022).
- Please include a short summary (can be tabulated) of the prior reviews on the investigated topic.
Answer: The authors added a tabulated summary of the previous review articles, trying to summarize the main context of these reviews in related fields of machine learning, informatics and manufacturing. Thus, the readers will benefit by a better understanding of the present work positioning in the machine learning and nanomanufacturing landscape.
Added: “So far, a lot of technical progress has been made in the field, as well as in docu-menting the recent progress and outcomes in the field. Below a table summary is pro-vided to present the landscape of reviews covering machine learning topics across dif-ferent domains/fields to better position the scope of this review study and the concepts covered in the following sections, which are connected to applications and implica-tions for nanomanufacturing.”
- At the end of the introduction section (or a separate section after the introduction), the aims and objectives of the review should be clearly described.
Answer: The authors revised the last paragraph of the introduction section to reflect the aims and objectives of the review more clearly.
Added: “This review focuses on the need to establish digitalized mapping between materi-als descriptors that arise from different characterization/simulation methods and re-late the chemical structure to nanomaterials options to favor the growth of dedicated properties profile and even to tune their production parameters to improve sustaina-bility. In section 2 the motivation is to introduce progress and challenges in discover-ing new materials by enabling shortcuts to rapid and safe-by-design routes, incorpo-rate experimental parameters in order to enhance decision-making and select the pro-cess parameters to produce nanomaterials to serve specific applications [40,43]. In ad-dition, in section 3 the improved capabilities for exploration of materials space and their corresponding digital representations are discussed and the key enabling steps to realize more flexibility in the production standards applied for selection of suitable nanomaterials, without the need to consider chemistry/physicochemistry or other complex theories in Materials Science, but using previous knowledge/experience doc-umented in corresponding datasets [40,45]. Another important mission of Machine In-telligence is in the field of the development of the safe and environmentally friendly nanomaterials, where in section 4 a threefold pattern in the prediction of their biologi-cal activity profile is presented based on data-driven representations, which could be prone to the availability of data and bias due to the applied procedures for sample preparation (contamination, purification), including biological systems containing human/algae cells, as well as in the bionanomaterial-cell interfaces. As presented in section 5, the foundation for this scope is data mining, management, and curation to enrich new or existing databases, as it is commonly accepted that the catalyst for AI is the accessibility to structured and Big Data [52,53]. Consequently, the existing methods can be supported by the computational theory that focuses on data, which holds a lot of promise to greatly reduce the computational and experimental pressure for innova-tion. At the end of this review the prospects of AI for the fields of Nanotechnology, Materials Science, and Nanoinformatics are summarized along with the conclusions of this study.”
- It is better to include a graphical abstract presenting the topics covered in the review for example names of the nanomaterials manufacturing technologies, the substrate materials, artifical intelligence/machine learning strategies ,and so on to cover the wide auience.
Answer: The graphical abstract has been revised. Names of nanomaterials, AI methods, and topics covered within the manuscript have been provided.
Added:
- The short-comings and potential applications of the listed/presented AI/ML methodologies should be highlighted. In particular, the physical descriptors (explainable artificial intelligence perspective/role should be discussed). The authors are suggested to get help from the following articles; "Boiling Heat Transfer Evaluation in Nanoporous Surface Coatings", "Liquid-to-vapor phase change heat transfer evaluation and parameter sensitivity analysis of nanoporous surface coatings", "On the assessment of the mechanical properties of additively manufactured lattice structures" and the like.
Answer: The authors elaborated on reviewer’s suggestion and added a corresponding paragraph under the section 3.
Added: “However, one main bottleneck commonly confronted is related to feature extrac-tion. Selecting the appropriate descriptors by implementing a Pearson parametric cor-relation map ensuring that informed predictions can be objective and trusted, towards the development of a model with unbiased establishment of parameters relation [100]. This is often a good strategy to avoid overfitting issues when developing machine learning models and improve the prediction accuracy, while excluding strongly corre-lated features [34]. Besides, bottlenecks related to computational resources and availa-bility of data/descriptors can be overcome based on the parametric sensitivity analysis and dependence plots indicates the parameters that are more influencing in the pre-dictions, thus selecting the most suitable descriptors for establishing multi-perspective predictive models [101,102].”
- The information presented in the discussion sections is better to be organized and in more readable/followable format.
Answer: The authors proceeded to minor revision to individual sections (2-5) to enhance readability. The discussion in the conclusion section has been thoroughly revised.
- The grammatical errors should be corrected.
Answer: The authors have proceeded to grammatical revisions, according to reviewer’s suggestion.
Round 2
Reviewer 2 Report
I accept the manuscript in the present form.
Reviewer 3 Report
The authors have revised the paper carefully, so it is accepted for publication.